# Structural basis of dynamic P5CS filaments

Jiale Zhong[†], Chen-Jun Guo[†], Xian Zhou[†], Chia-Chun Chang, Boqi Yin, Tianyi Zhang, Huan-Huan Hu, Guang-Ming Lu, Ji-Long Liu*

School of Life Science and Technology, ShanghaiTech University, Shanghai, China

**Abstract** The bifunctional enzyme Δ[1]-pyrroline-5-carboxylate synthase (P5CS) is vital to the synthesis of proline and ornithine, playing an essential role in human health and agriculture. Pathogenic mutations in the P5CS gene (ALDH18A1) lead to neurocutaneous syndrome and skin relaxation connective tissue disease in humans, and P5CS deficiency seriously damages the ability to resist adversity in plants. We have recently found that P5CS forms cytoophidia in vivo and filaments in vitro. However, it is difficult to appreciate the function of P5CS filamentation without precise structures. Using cryo-electron microscopy, here we solve the structures of *Drosophila* full-length P5CS in three states at resolution from 3.1 to 4.3 Å. We observe distinct ligand-binding states and conformational changes for the GK and GPR domains, respectively. Divergent helical filaments are assembled by P5CS tetramers and stabilized by multiple interfaces. Point mutations disturbing those interfaces prevent P5CS filamentation and greatly reduce the enzymatic activity. Our findings reveal that filamentation is crucial for the coordination between the GK and GPR domains, providing a structural basis for the catalytic function of P5CS filaments.

## Editor's evaluation

This paper reports the cryo-EM structures of *Drosophila* P5CS, an enzyme important in amino acid metabolism. This group had previously described P5CS filaments in *Drosophila*, and here show how the filaments are assembled. The paper describes structural changes that occur upon the binding of substrates and reaction intermediates, making a strong case for a conformational cycle that involves some loop movements. Importantly, the work shows that these movements occur in the context of the assembled filament. Point mutants that block filament assembly have reduced catalytic rates, suggesting that a role of the filament is to increase enzyme activity.

*For correspondence:
liujl3@shanghaitech.edu.cn

[†]These authors contributed equally to this work

**Competing interest:** The authors declare that no competing interests exist.

## Introduction

The bifunctional enzyme Δ[1]-pyrroline-5-carboxylate synthase (P5CS) is responsible for proline and ornithine metabolism (*Baumgartner et al., 2000*; *Baumgartner et al., 2005*; *Hu et al., 2008*; *Pérez-Arellano et al., 2010*). In humans, over 30 mutations in P5CS have been identified as the causes of rare diseases (*Baumgartner et al., 2000*; *Baumgartner et al., 2005*; *Marco-Marín et al., 2020*; *Pérez-Arellano et al., 2010*; *Skidmore et al., 2011*). In addition, the glutamine-proline regulatory axis has been considered a promising target for cancer therapy (*Guo et al., 2020*; *Liu et al., 2012*). In plants, proline synthesis is associated with plant stress resistance (*Pérez-Arellano et al., 2010*). Therefore, P5CS is of great significance in human health and agriculture.

Previous studies have revealed a characteristic compartmentation of enzymes via filamentation (*Hunkeler et al., 2018*; *Johnson and Kollman, 2020*; *Liu, 2010*; *Park and Horton, 2019*; *Stoddard et al., 2020*). This filamentous structure is membraneless and termed the cytoophidium for its appearance (*Liu, 2010*; *Liu, 2016*). The cytoophidium has emerged as a mechanism for the regulation of

metabolic enzymes (*Hansen et al., 2021*; *Liu, 2016*; *Zhou et al., 2021*). Recently, we have shown that *Drosophila* P5CS forms cytoophidia in vivo and forms individual filaments in vitro (*Zhang et al., 2020*).

P5CS corresponds to two individual proteins in prokaryotes and some lower eukaryotes such as yeast. One is the glutamate kinase (GK, *pro*B gene), and the other is γ-glutamyl phosphate reductase (GPR, *pro*A gene) in *Escherichia coli*. Kinetic analysis suggest that bacterial GK and GPR form a complex (*Gamper and Moses, 1974*). The dual functions of P5CS in higher eukaryotes implicate that both GK and GPR have evolved into one single protein for coupling reactions. However, no structure of the full-length P5CS has been solved. The underlying mechanisms of the catalytic reaction and the function of filamentation remain unknown.

Using cryo-electron microscopy (cryo-EM), here we solve the structures of full-length P5CS in multiple filamentous states. We reconstruct *Drosophila* P5CS structures at 3.1–4.3 Å resolutions, providing detailed information of the P5CS filaments bound with different ligands. Our results describe the assembly mechanism of P5CS filaments, in which the GK domain forms tetramer and the GPR domain forms dimer structure, and both domains form specific interaction interfaces. Based on these structures, we propose a working model that filamentation is critical for the coordinated reactions between GK and GPR, the two domains of P5CS.

## Results

### Overall structures of P5CS filaments

The P5CS molecule contains two domains, GK and GPR, catalyzing the first and second steps in the biosynthesis of proline from glutamate. The GK domain catalyzes glutamate phosphorylation, and the GPR domain catalyzes the NADPH-dependent reduction of γ-glutamyl phosphate (G5P) to glutamate-γ-semialdehyde (GSA). The end product P5C, formed by a spontaneous cyclization reaction of GSA (*Figure 1A*), will be used by another enzyme P5C reductase (P5CR) to produce proline.

In order to solve the structure of P5CS filaments, we expressed and purified *Drosophila melanogaster* full-length P5CS proteins. First, we analyzed the APO and substrate-bound states of P5CS by negative staining (*Figure 1—figure supplement 1A–D*). In our previous study, we found that *Drosophila* P5CS in the APO state is hard to form filaments at low concentrations (<0.05 μM). The addition of glutamate to the P5CS samples induces micron-scale filaments (*Zhang et al., 2020*). Here, we observe that increasing P5CS concentration (>1 μM) also promotes the formation of filaments in the APO state. Our results show that the P5CS proteins can be self-assembled into filaments without ligands, and adding all substrates increases the length of filaments at the same concentration of the P5CS proteins. Consistent with our previous study, glutamate (a substrate of P5CS) promotes the formation and maintenance of *Drosophila* P5CS filaments (*Zhang et al., 2020*).

Subsequently, samples of the P5CS proteins incubated with different combinations of substrates were prepared for cryo-EM (*Figure 1—figure supplement 1E–J*). Filaments in three conditions with (1) glutamate (P5CS$^{Glu}$), (2) glutamate and ATPγS (P5CS$^{Glu/ATPγS}$), and (3) glutamate, ATP, and NADPH (P5CS$^{Mix}$) were imaged in cryo-EM for single-particle analysis (SPA). Long and flexible filaments of P5CS were observed under all the three conditions. After 3D classification and 3D reconstruction, the electron density maps of the P5CS$^{Glu}$, P5CS$^{Glu/ATPγS}$, and P5CS$^{Mix}$ filaments reached resolutions of 4.0 Å, 4.2 Å, and 3.6 Å, respectively (*Figure 1B–D*, *Figure 1—figure supplements 2–4*). Using a separate focused refinement strategy, we obtained multiple conformational states of the GK domain tetramer (3.1–3.5 Å) and the GPR domain dimer (3.6–4. 3Å). The cryo-EM data and model refinement statistics are provided in *Table 1*. The N-terminus (residues 1–44) and three disordered segments in regions I, II, and III in the GK domain were invisible in our maps.

One P5CS monomer can be roughly divided into five subdomains: (1) the glutamate-binding domain (GBD) and (2) the ATP-binding domain (ABD) at the GK domain; (3) the NADPH-binding domain (NBD), (4) the catalytic domain (CD), and (5) the oligomerization domain (OD) at the GPR domain (*Figure 1A and E*). In the model, two P5CS monomers dimerize through the interaction between their GPR domains, where the β21 at the CD interacts with the β24 at the OD of the other monomer (*Figure 1F*). This interaction connects two groups of hairpins and maintains the homodimer structure by a hydrogen bond network. Two P5CS dimers further assemble into a compact tetramer through the interaction at the GK domains. The P5CS tetramer serves as the building block of P5CS filaments (*Figure 1G*).

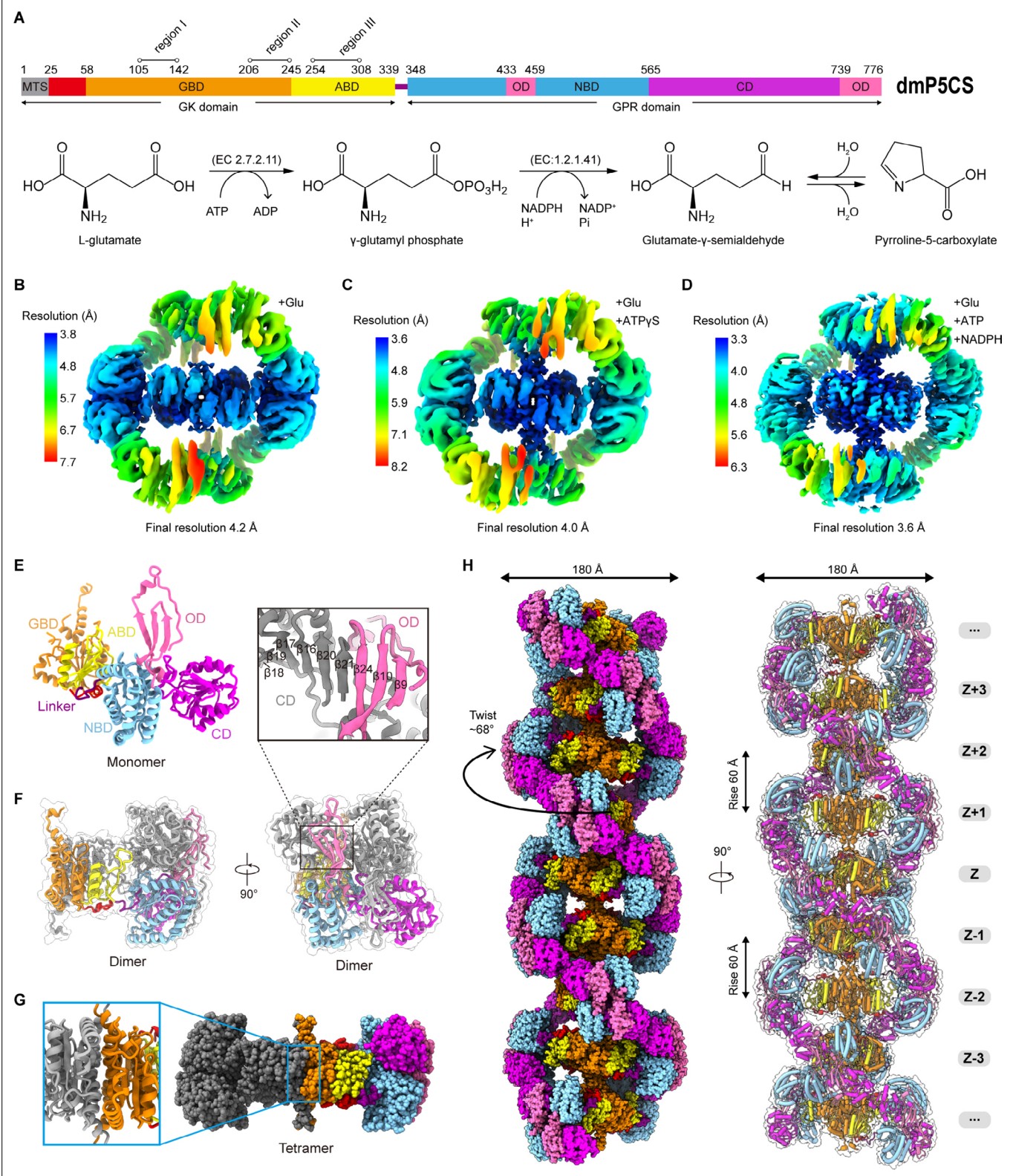

**Figure 1.** Bifunctional enzyme properties and cryo-electron microscopy (cryo-EM) analysis of P5CS filaments. (**A**) Domain organization of *Drosophila melanogaster* P5CS, which consists of two domains, N-terminal glutamate kinase (GK) domain and C-terminal γ-glutamyl phosphate reductase (GPR) domain. Putative mitochondrial targeting sequence (MTS) is labeled in gray; the glutamate-binding domain (GBD) and the ATP-binding domain (ABD) of the GK domain are respectively shown in orange and yellow; the NADPH-binding domain (NBD), the catalytic domain (CD), and the oligomerization

*Figure 1 continued on next page*

*Figure 1 continued*

domain (OD) of the GPR domain are shown in cyan, purple, and pink, respectively. Bifunctional P5CS enzyme catalytic reaction and residue numbers for domain boundaries are shown. (**B–D**) Single-particle analysis for 3D reconstruction of P5CS filaments, three cryo-EM maps of P5CS^Glu filament, P5CS^Glu/ATPγS filament, and P5CS^Mix filament are colored by local resolution estimations. (**E**) The structures of the P5CS monomer and color codes for P5CS models are indicated. (**F**) The P5CS dimer. Two monomers (gray or color coded by domain) interact via GPR domain hairpins contact. (**G**) The P5CS tetramer (sphere representation) is formed via GK domain interaction (cartoon representation) between two P5CS dimers (gray or color coded by domain). (**H**) The sphere and cartoon representation of P5CS filaments. P5CS filaments are modeled by the cryo-EM map. The rotated view is shown in the right panel; its rise, twist, and width are indicated.

The online version of this article includes the following video and figure supplement(s) for figure 1:

**Figure supplement 1.** Substrates can significantly extend the P5CS filament.

**Figure supplement 2.** Cryo-electron microscopy (cryo-EM) analysis of the P5CS^Mix filament.

**Figure supplement 3.** Quality of cryo-electron microscopy (cryo-EM) maps.

**Figure supplement 4.** Representative cryo-electron microscopy (cryo-EM) map.

**Figure 1—video 1.** Morph between the consensus structures of P5CS^Glu filaments.

https://elifesciences.org/articles/76107/figures#fig1video1

**Figure 1—video 2.** Morph between the consensus structures of P5CS^Mix filaments.

https://elifesciences.org/articles/76107/figures#fig1video2

---

P5CS filament structures in all the three states showed characteristics of double helix (*Figure 1—video 1*, *Figure 1—video 2*). We chose the P5CS^Mix filament to display the details (*Figure 1H*). In the helical P5CS filament structure, the GK tetramers serve as the core of the filament, and the GPR dimers form left-handed double helix structures around the central axis. The overall diameter of P5CS filaments in all three states is 180 Å, while the helical twist is 68° and the helical rise is 60 Å (*Figure 1H*).

## Structural comparison of ligand-bound GK domains

The GK domain of *Drosophila* P5CS is conserved with the GK protein in *E. coli*. Alignments of sequences and structures indicate that their secondary structures are similar as both exhibit a sandwich-like α3β8α4 topological folding (*Figure 2—figure supplement 1A*), which is a characteristic of the amino acid kinase (AAK) family (*Marco-Marín et al., 2007*; *Pérez-Arellano et al., 2010*; *Ramón-Maiques et al., 2002*).

We obtained a structure of the GK domain with the binding of glutamate in the P5CS^Glu filament (*Figure 2A*) and a second structure of the GK domain with G5P-Mg-ADP in the P5CS^Mix filament (*Figure 2B*, *Figure 2—figure supplement 1B*). In the P5CS^Glu/ATPγS filament, the ligands could not be determined due to incomplete densities (*Figure 2—figure supplement 1C*). The GK domain structure of the P5CS^Glu/ATPγS filament is virtually identical to that of the P5CS^Mix filament (*Figure 2—figure supplement 1D*). We speculate that there are two ligand-binding modes (bound with Glu-Mg-ATPγS and G5P-Mg-ADP, respectively) in the P5CS^Glu/ATPγS filament. These two modes may coexist in the active sites of the GK tetramer, thereby affecting the 3D reconstruction of the structures. The unexpected presence of G5P could be due to the contamination of ATP in the commercial ATPγS (80% pure) and all substrates were in excess during our sample preparation. Thus, no ligand was modeled in the GK domain structure of the P5CS^Glu/ATPγS filament.

In the GK domain, a valley-like pocket locates between GBD and ABD, providing the binding sites for glutamate, ATP, or their derivatives (*Figure 2C and D*). Glutamate binds to the active site of GBD vertically (*Figure 2A and C*, *Figure 2—figure supplement 1E*). In contrast, G5P and ADP extend towards each other in the P5CS^Mix filament, and glutamate at the binding site is converted into the intermediate G5P. At ABD of the P5CS^Mix filament, the phosphate donor ATP becomes an ADP, associating with an $Mg^{2+}$ (*Figure 2B and D*).

Superimposing the GK tetramer in the P5CS^Glu filament and that in the P5CS^Mix filament revealed that the major motion of the GK domain occurred at the region containing flexible loops or disordered segments, whereas the α3β8α4 fold showed minor movement (*Figure 3A*). Meanwhile, based on the disorder densities in region II (*Figure 3—figure supplement 1A–D*), we modeled the possible trend of the missing segment with a dashed line (*Figure 3A*). In the P5CS^Glu filament, we speculate that the disordered segment in region II acts as a closed loop, which traps glutamate in GBD (*Figures 2C and 3A*). In the P5CS^Mix filament, the same segment shifts away from the top of the binding pocket and

**Table 1.** Cryo-electron microscopy (cryo-EM) data statistics.

|  | P5CS$^{Glu}$ filament | P5CS$^{Glu/ATP\gamma S}$ filament | P5CS$^{Mix}$ filament |
|---|---|---|---|
| **Data collection and processing** | | | |
| EM equipment | Titan Krios | Titan Krios | Titan Krios |
| Detector | K3 camera | K3 camera | K3 camera |
| Magnification | 22,500× | 22,500× | 22,500× |
| Voltage (kV) | 300 | 300 | 300 |
| Electron exposure (e–/Å²) | 72 | 72 | 72 |
| Defocus range (µm) | –0.8 to –2.5 | –0.8 to –2.5 | –0.8 to –2.5 |
| Pixel size (Å) | 0.53 | 0.53 | 0.53 |
| Symmetry imposed | D2 | D2 | D2 |
| Number of collected movies | 4933 | 6408 | 10,566 |
| Initial particle images (no.) | 1,911,843 | 1,563,553 | 8,027,582 |
| Final particle images (no.) | 432,746 | 327,841 | 1,412,498 |

**Refinement**

|  | P5CS tetramer | GK domain | GPR domain | P5CS tetramer | GK domain | GPR domain | P5CS tetramer | GK domain | GPR domain closed form | GPR domain open form |
|---|---|---|---|---|---|---|---|---|---|---|
| EMDB ID | EMD-31466 | EMD-31469 | EMD-32877 | EMD-31467 | EMD-32876 | EMD-32880 | EMD-31468 | EMD-32875 | EMD-32878 | EMD-32879 |
| PDB code | 7F5T | 7F5X | 7WXF | 7F5U | 7WX4 | 7WXI | 7F5V | 7WX3 | 7WXG | 7WXH |
| Initial model used (PDB code) | - | 4Q1T | 2H5G | - | 4Q1T | 2H5G | - | 4Q1T | 2H5G | 2H5G |
| Map resolution (Å) | 4.1 | 3.5 | 3.6 | 4.1 | 3.4 | 4.2 | 3.6 | 3.1 | 4.2 | 4.3 |
| FSC threshold | 0.143 | 0.143 | 0.143 | 0.143 | 0.143 | 0.143 | 0.143 | 0.143 | 0.143 | 0.143 |
| Map resolution range (Å) | 3.8–8.0 | 3.4–5.2 | 3.5–5.0 | 3.4–8.0 | 3.2–4.7 | 4.1–5.3 | 3.3–7.8 | 3.0–4.1 | 4.1–5.9 | 4.0–5.5 |
| Map sharpening B-factor (Å²) | –120 | –120 | –120 | –100 | –70 | –200 | –80 | –80 | –150 | –150 |
| *Model composition* | | | | | | | | | | |
| Non-hydrogen atoms | 20,436 | 7244 | 6,494 | 20,744 | 7968 | 6522 | 20,912 | 8172 | 6494 | 6590 |
| Protein residues | 2700 | 1896 | 860 | 2740 | 1040 | 860 | 2760 | 1064 | 860 | 430 |
| Ligands | GGL | GGL | - | - | - | RGP | - | RGP, ADP | NAP | - |
| Ions | 0 | 0 | 0 | 0 | 0 | 0 | 0 | Mg | 0 | 0 |
| *B factors (Å²)* | | | | | | | | | | |
| Protein | 140 | 150 | 162 | 143 | 74 | 121 | 131 | 62 | 123 | 100 |
| Ligand | 140 | 150 | - | - | - | 145 | - | - | - | 121 |
| *R.m.s. deviations* | | | | | | | | | | |

*Table 1 continued on next page*

*Table 1 continued*

|  | P5CS^Glu filament | | | P5CS^Glu/ATPγS filament | | | P5CS^Mix filament | | | |
|---|---|---|---|---|---|---|---|---|---|---|
| Bond lengths (Å) | 0.005 | 0.005 | 0.007 | 0.007 | 0.006 | 0.007 | 0.005 | 0.006 | 0.008 | 0.005 |
| Bond angles (°) | 0.678 | 0.54 | 0.777 | 0.786 | 0.576 | 0.788 | 0.675 | 0.554 | 0.864 | 0.73 |
| *Validation* | | | | | | | | | | |
| MolProbity score | 2.73 | 2.53 | 2.25 | 2.47 | 2.51 | 2.59 | 2.17 | 1.85 | 2.89 | 2.19 |
| Clashscore | 47.48 | 7.93 | 13.17 | 23.56 | 8.2 | 24.57 | 16.57 | 5.45 | 34.6 | 12.87 |
| Poor rotamers (%) | 0 | 6.12 | 0.29 | 0 | 6.39 | 0.58 | 0.18 | 3.56 | 1.74 | 0.29 |
| *Ramachandran plot* | | | | | | | | | | |
| Favored (%) | 89.04 | 91.67 | 87.15 | 87.59 | 92.8 | 81.78 | 92.96 | 97.27 | 93.29 | 89.25 |
| Allowed (%) | 10.51 | 7.89 | 12.62 | 12.26 | 7.2 | 18.22 | 6.89 | 2.73 | 16.71 | 10.75 |
| Disallowed (%) | 0.45 | 0.44 | 0.23 | 0.15 | 0 | 0.58 | 0.15 | 0 | 0 | 0 |

GPR: γ-glutamyl phosphate reductase; GK: glutamate kinase; FSC: Fourier shell correlation.

forms an open loop, in which residue M213 interacts with G5P (*Figure 3A*, *Figure 3—figure supplement 1E*). We notice that the closed loop has a steric clash with G5P, preventing the binding of G5P under such a conformation (*Figure 3—figure supplement 1D*). Our findings support the idea that region II at the GK domain engages in regulating the catalytic reaction.

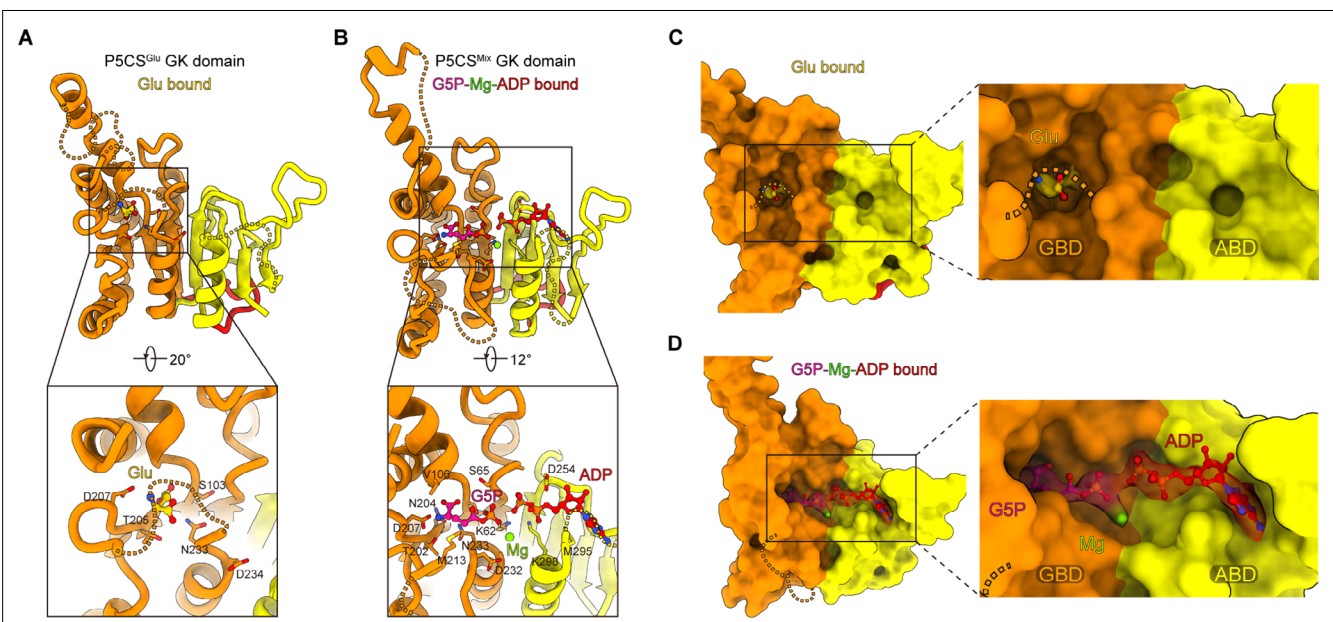

**Figure 2.** Conformational changes in the glutamate kinase (GK) domain-binding pocket. (**A**) GK domain of the P5CS^Glu filament, with glutamate shown as sticks with yellow carbons. The dashed lines represent disordered segments (residues 124–142, 211–232, and 275–297) in this model. (**B**) GK domain of the P5CS^Mix filament, with G5P, Mg^+, and ADP shown as sticks with pink, green, and red carbons, respectively. The dashed lines represent disordered segments (residues 128–140, 214–228, and 282–295) in this model. (**C, D**) GK domain model surface representation showing the conformation of the binding pocket in the P5CS^Glu filament or P5CS^Mix filament. The cryo-electron microscopy (cryo-EM) density of binding glutamate molecule in (**C**), and the binding complex of G5P, Mg^+, and ADP in (**D**). The dashed lines represent 'open loop' and 'closed loop'

The online version of this article includes the following figure supplement(s) for figure 2:

**Figure supplement 1.** Structural details of the glutamate kinase (GK) domain characterized.

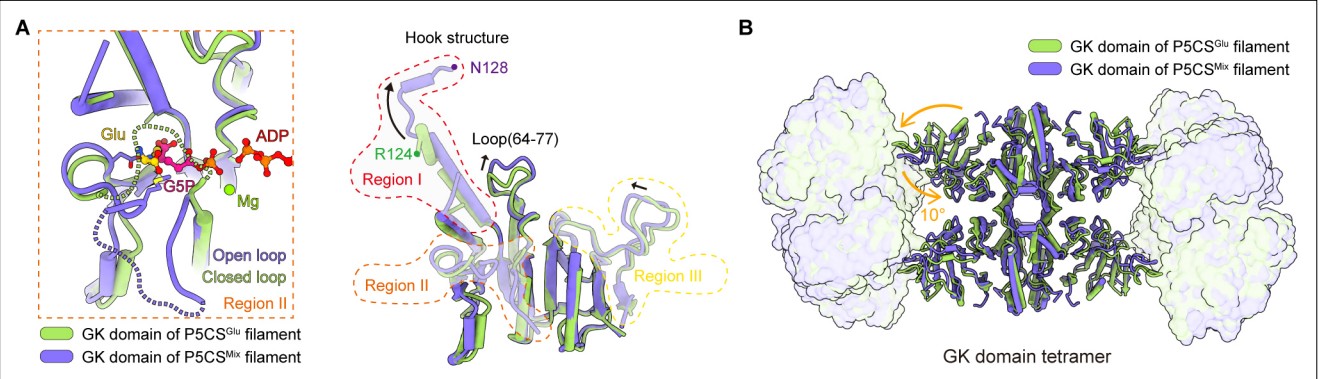

**Figure 3.** Structural comparison of the two types of glutamate kinase (GK) domain. (**A**) Comparison of one protomer of the GK domain tetramer in the P5CS[Glu] filament (green) and P5CS[Mix] filament (blue-violet) on the right panel. On the left panel, the dashed lines in the model represent the open loop (blue-violet) and closed loop (green) in region II. (**B**) Superimposition of the GK domain tetramer in the P5CS[glu] filament (green) with the P5CS[Mix] filament (blue-violet). Transitions from glutamate-bound-conformation to G5P-Mg-ADP-bound conformation are shown as curved arrows, indicating GK domain conformational changes in the P5CS filament.

The online version of this article includes the following figure supplement(s) for figure 3:

**Figure supplement 1.** The conformational changes in regions I and II.

The helix-helix structure (residues 105–113, 115–124) at region I of the P5CS[Glu] filament transforms into a helix-loop-helix structure (residues 105–119, 120–122, 123–128) in the P5CS[Mix] filament (*Figure 3A*, *Figure 3—figure supplement 1F*). This helix-loop-helix structure is referred to as the 'hook' structure. The transformation of the hook structure results in new contact sites between neighbor tetramers in the vertical direction, which is evidenced by a rigid density in our map (*Figure 1B–D*, *Figure 1—figure supplements 2 and 3*).

On the other hand, we notice the conformational variation of the loop at region III and a loop (residues 64–77) of GBD shifting greatly by approximately 3 Å away (*Figure 3A*). The function of these conformational changes is unclear, which may relate to conformational changes of the active site. In order to investigate the conformational changes involved in the catalytic reaction, we further compared the tetramer structures of the GK domain in the P5CS[Glu] and P5CS[Mix] filaments (*Figure 3B*). The GK domain of each protomer rotates approximately 10° around its central axis, causing the horizontal compression of the GK domain dimer. By comparing the structures of the GK domain with various ligands, we demonstrate the conformational changes, which may be associated with phosphorylation of the substrate glutamate.

## Open and closed conformations of GPR domains

The GPR domain of P5CS belongs to the aldehyde dehydrogenase (ALDH) superfamily. ALDH family uses NAD(P)$^+$ to catalyze the conversion of various aldehydes into their corresponding carboxylic acids. Many studies on ALDHs have shown that a conserved residue cysteine acts as the active site of nucleophile, forming thiohemiacetal intermediate with substrate (*Koppaka et al., 2012*; *Liu et al., 1997*; *Perozich et al., 1999*). Curiously, the NADPH-utilizing GPR domain of P5CS catalyzes the reverse reaction of ALDHs.

On the basis of P5CS structures, we display four different binding modes (*Figure 4A–D*) of the GPR domain. In the P5CS[Glu] filament, no ligand binds to the GPR domain (*Figure 4A*). In the P5CS[Glu/ATPγS], however, we observed the density of a G5P at the CD active site (*Figure 4B*, *Figure 4—figure supplement 1A*). It might be a contamination of ATP, leading to the production of the substrate G5P. In this model, the binding mode of G5P (referred to as the G5P-binding state) is clearly solved. By focus refinement of the GPR dimer in the P5CS[Mix] filament, we determined two additional states of the GPR domain (*Figure 4C and D*). One is the NADP(H)-binding state, when NADP(H) is present at NBD (*Figure 4—figure supplement 1B*). The other is the NADP(H)-released state, of which the cofactor binding site is empty.

Conformational comparison of unliganded, G5P-binding and NADP(H)-released states shows that the overall structures of the GPR domain are similar (*Figure 4—figure supplement 2A*). The structure

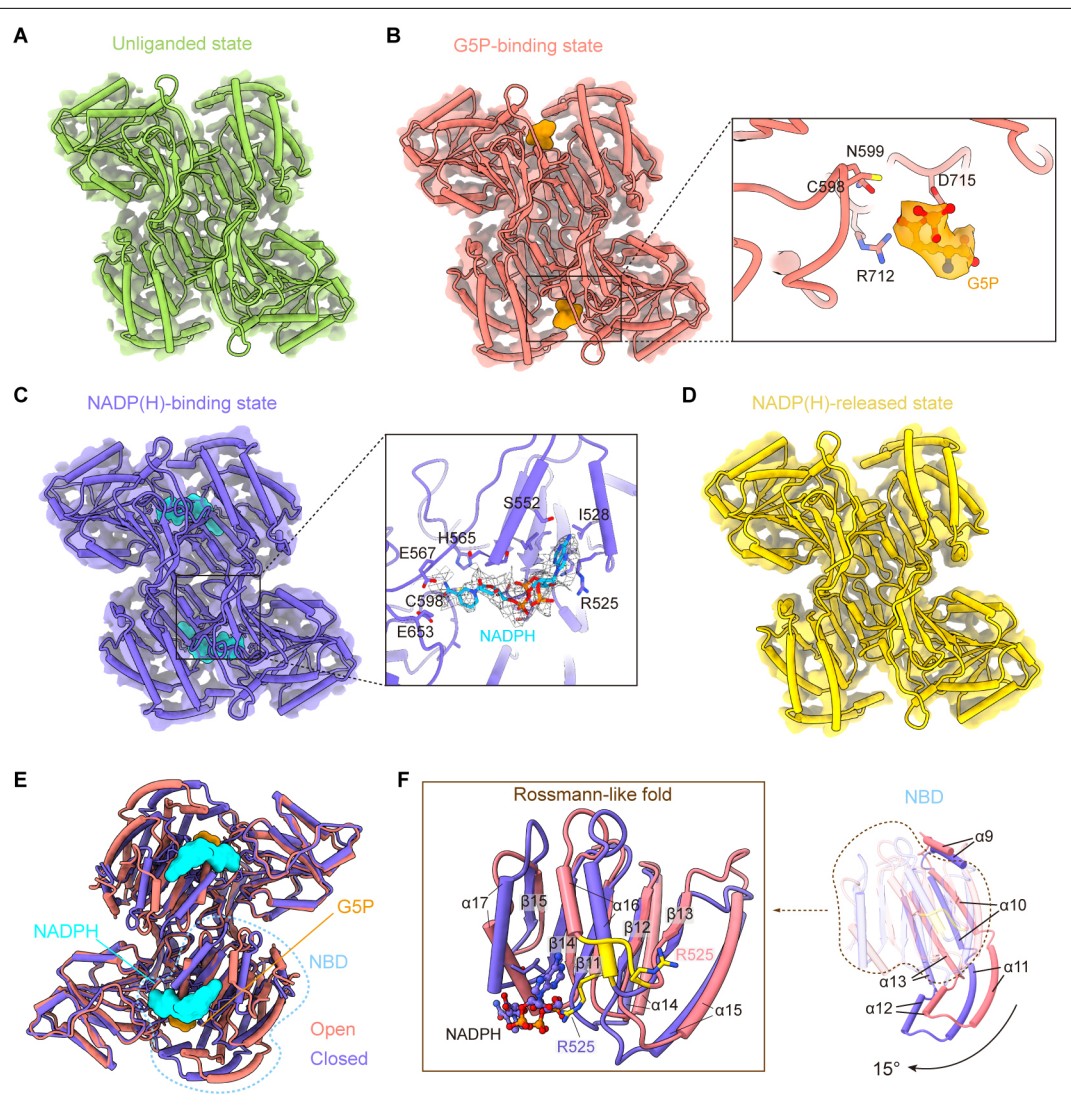

**Figure 4.** γ-Glutamyl phosphate reductase (GPR) domain ligand-bound mode and its conformation. (**A**) The cryo-electron microscopy (cryo-EM) density of the GPR dimer structure and cartoon model is represented as an unliganded state in the P5CS[Glu] filament (green). (**B**) GPR dimer structure of the G5P-binding state in the P5CS[Glu/ATPγS] filament (coral). The conformation of the G5P-binding pocket and G5P (orange) is shown as stick representation. (**C**) GPR dimer structure of the NADP(H)-binding state in the P5CS[Mix] filament (blue-violet). The conformation of the NADP(H)-binding pocket with NADPH (cyan) is shown as stick representation. (**D**) GPR dimer structure of the NADP(H)-released state in the P5CS[Mix] filament (yellow). (**E**) Structural differences in the G5P-binding state (coral) and NADP(H)-binding state (blue-violet) of the GPR domain. Ligands are colored as in (**B, C**). (**F**) Superimposition of either the NADPH-binding domain (NBD) or the Rossmann-fold of the GPR domain at the G5P-binding state and NADP(H)-binding state using a single protomer.

The online version of this article includes the following video and figure supplement(s) for figure 4:

**Figure supplement 1.** Representative cryo-electron microscopy (cryo-EM) densities for the active site of the γ-glutamyl phosphate reductase (GPR) domain.

**Figure supplement 2.** Comparison of the structures of the γ-glutamyl phosphate reductase (GPR) domain.

**Figure supplement 3.** The NADPH-binding domain (NBD) rotation and view of the active site of the γ-glutamyl phosphate reductase (GPR) domain with its substrate.

**Figure 4—video 1.** Structural transition of open and closed conformations of the γ-glutamyl phosphate reductase (GPR) domain.

https://elifesciences.org/articles/76107/figures#fig4video1

of the NADP(H)-release state, which has no bound ligand, is identical to the unliganded state. The binding of G5P to CD of the GPR domain does not lead to obvious conformational changes.

Next, we compared the structures of the G5P-binding state and NADPH-binding state (*Figure 4E*, *Figure 4—figure supplement 2B*). We have found that the structures of CD and OD are generally consistent in those two states, while NBD of the GPR domain in those two states differs greatly (*Figure 4E*). NBD contains consecutive alternating α-helices and β-strands ($\alpha_2$-$\beta_5$-$\alpha_2$) architecture, which is known as the Rossmann-like fold for dinucleotide binding (*Cheek et al., 2005*). By superimposing the Rossmann-like fold and the entire NBD, we determined conformational changes between the GPR domain at the G5P-binding state and that at the NADP(H)-binding state (*Figure 4F*).

Upon NADP(H) binding, the residue R525 interacts with the adenine moiety. This interaction transforms the $^{525}REE^{527}$ loop into an ordered structure that extends the α16 helix in the Rossmann-like fold (*Figure 4F*). Meanwhile, the entire NBD rotates approximately 15° along the cylinder axis (*Figure 4—figure supplement 3A*) and slides towards CD (*Figure 4F*; *Figure 4—video 1*). We hypothesize that the helix, when turns disordered, loses contact with the adenine moiety and then separates the cofactor from NBD via a conformational selection mechanism. A similar phenomenon was also observed in ALDH1L1 (*Tsybovsky and Krupenko, 2011*). This transformation contributes to bringing the nicotinamide ring of the NADP(H) close to the catalytic residue C598 of CD. Conformational changes triggered by the binding of NADPH subsequently initiate the transfer of the hydride ion from NADPH to the intermediate G5P (*Figure 4—figure supplement 3B*).

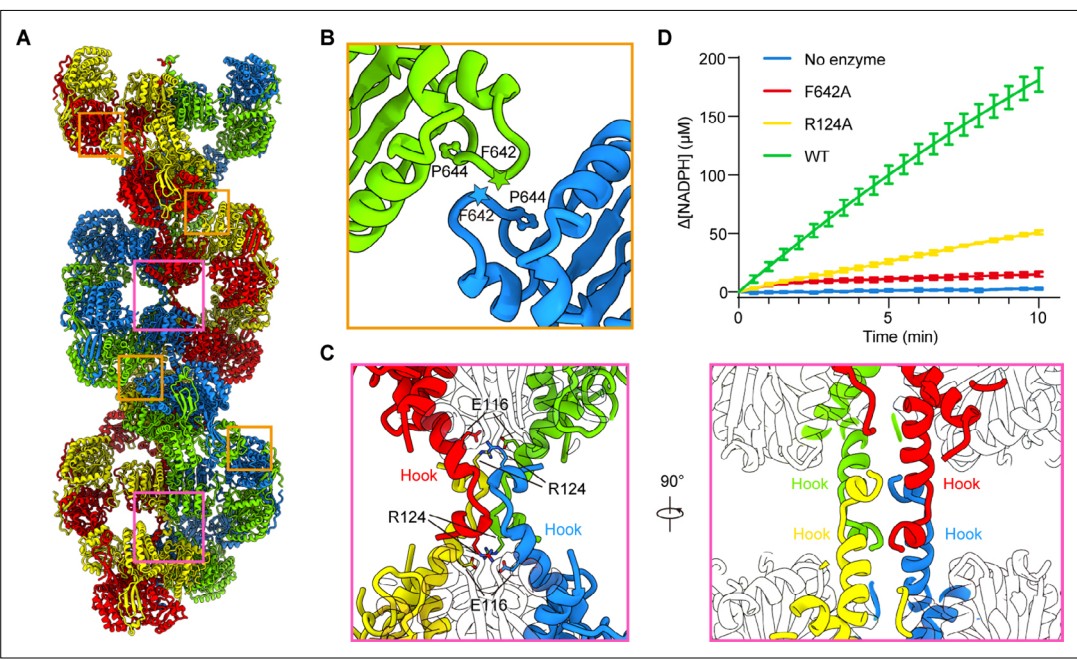

**Figure 5.** Assembly and interaction surfaces of the P5CS filament. (**A**) P5CS filament assembly interface, the four P5CS protomers in one layer are colored in red, yellow, blue, and green. (**B**) Interaction between two adjacent γ-glutamyl phosphate reductase (GPR) domain dimers, residues F642 located at loop that interacts with P644 from another neighboring GPR domain dimer. (**C**) Model for hook structure interaction. (**D**) Enzyme activity analysis to examine P5CS wild-type or mutant proteins. All of the experiments were replicated three times (n = 3, mean ± SD).

The online version of this article includes the following video, source data, and figure supplement(s) for figure 5:

**Source data 1.** Enzymatic activity of wild-type and mutant *Drosophila* P5CS.

**Figure supplement 1.** The interface of adjacent γ-glutamyl phosphate reductase (GPR) domain dimers.

**Figure supplement 2.** Negative staining of mutated P5CS.

**Figure supplement 3.** The distance between the active sites of the glutamate kinase (GK) domain and γ-glutamyl phosphate reductase (GPR) domain.

**Figure supplement 4.** Sequence alignment of the representative P5CS enzymes.

**Figure 5—video 1.** Simulated ligand-binding site of P5CS filament.

https://elifesciences.org/articles/76107/figures#fig5video1

In our models, unliganded, G5P-binding and NADP(H)-released states represent as open conformation, and the NADP(H)-binding state represents as closed conformation. We propose that the P5CS filament accommodates the GPR domain in both open and closed conformations. Therefore, recurring transformations between these two conformations are essential for the catalytic cycle at the GPR domain.

## Filamentation regulates the enzymatic reaction

In P5CS$^{Glu}$, P5CS$^{Glu/ATPγS}$, and P5CS$^{Mix}$ filaments, neighboring GPR dimers interact with each other and form a helical structure. The interaction formed between F642-P644 of the contact loops in adjacent GPR domains, which appears as a CH/Pi interaction (*Zondlo, 2013*), is critical for the filamentation (*Figure 5A and B*, *Figure 5—figure supplement 1*). In P5CS$^{Glu/ATPγS}$ and P5CS$^{Mix}$ filaments, the additional interface between hook structures pairs locks adjacent GK tetramers. The hook structure extrudes from the GK domain. In a GK tetramer, four hooks extrude toward two opposite directions to form a 'spinning top' arrangement. Therefore, two pairs of hooks in a GK tetramer interact with their counterparts in two adjacent GK tetramers (*Figure 5A and C*). The hook interaction forms strong contacts via hydrogen bonds (M119-R124, L121-M123) and salt bridges (E116-R124). Therefore, a combination of the GPR contact (for the double helix) and GK contact (for the axis) stabilizes the P5CS filament.

To understand the function of P5CS filaments, we generated two mutants, R124A and F642A, which are predicted to abrogate the tetramer-tetramer contact sites of the GK domains and GPR domains, respectively. Negative stain of the mutant P5CS showed that the P5CS$^{F642A}$ mutant proteins did not assemble into a filament with or without ligands (*Figure 5—figure supplement 2A*). These results indicate that the interaction at the GPR domain interface is crucial for P5CS filamentation.

In contrast, the P5CS$^{R124A}$ mutant proteins formed long filaments in the APO state as well as in the presence glutamate (*Figure 5—figure supplement 2B*). We observed that glutamate-bound P5CS$^{R124A}$ filaments disassembled at the initial phase of adding ATP. Being incubated with all substrates, P5CS$^{R124A}$ formed shorter filaments than P5CS$^{WT}$ (*Figure 5—figure supplement 2B*).

We propose that the interactions among the hook pairs are required for stabilizing the filament during the transformation from the P5CS$^{Glu}$ filament to the P5CS$^{Glu/ATPγS}$ filament or P5CS$^{Mix}$ filament. We subsequently analyzed the activity of the wild-type P5CS and two mutants, R124A and F642A. The two mutants exhibited a dramatically compromised activity in comparison with the wild-type P5CS (*Figure 5D*), suggesting that the integrity of filament is essential to the catalytic reactions.

## Discussion

### The GK domain

We observed two ligand-binding modes in the GK domain. Due to the lack of ATP-bound structure, it is difficult to determine whether ATP plays a decisive role in these conformational changes. According to a previous report on the N-acetyl-L-glutamate kinase (NAGK), nucleoside is important for the conformational change of the AAK domain, and the structures are similar when bound by ADP or AMPPNP (*Gil-Ortiz et al., 2011*). Based on the similarity of sequences and structures between GK and NAGK (*Marco-Marín et al., 2007*), we propose that the conformation of the GK domain in the P5CS$^{Glu}$ filament would transform upon the binding of ATP, thereby triggering the formation of hook structure and completing the catalytic reaction. Although we solved the clear structure of the P5CS$^{Glu}$ filament, further research is needed to understand how the conformation of glutamate binding contributes to the extension of P5CS filaments.

### The GPR domain

Aspartate-β-semialdehyde dehydrogenase (ASADH) catalyzes NADPH-dependent reductive dephosphorylation of β-aspartyl phosphate to aspartate-β-semialdehyde (*Karsten and Viola, 1991*). The GPR domain of P5CS and ASADH catalyzes the same type of reaction. Interestingly, the binding of NADP$^+$ will change the cofactor binding domain of ASADH from open conformation to closed conformation (*Hadfield et al., 2001*). Thus, we speculate that their catalytic mechanisms have something in common. In the GPR domain of *Drosophila* P5CS, our data suggest that the catalytic residue C598 of CD attacks the G5P to form the first tetrahedral thioacetal intermediate in the reaction, and then

expulsion of phosphate collapses to form a stable thioacyl enzyme intermediate. A hydride is then transferred to this intermediate from NADPH, with subsequent collapse to release the product GSA. Furthermore, the NADPH-binding site is located inside the filament, close to the GK domain. The G5P binding site is close to the external solution environment, which is proposed to facilitate the release of the product GSA/P5C (*Figure 5—figure supplement 3*, *Figure 5—video 1*). G5P can freely bind to the GPR domain in our model (G5P-binding state). However, in the closed conformation, when the nicotinamide ring of NADP(H) approaches the G5P-binding site, the substrate tunnel entrance may be blocked by NADP(H). This may affect the subsequent binding of G5P. Therefore, we speculate that the GPR domain should bind with G5P prior to NADPH binding. However, whether this mechanism is a preferred binding order needs to be further verified by kinetic experiments.

## The P5CS filament

As mentioned in the 'Results' section, we observed that mutated residues R124A and F642A do not directly participate in the active sites, while they are crucial for filamentation. This suggests that the P5CS filamentation couples the reaction catalyzed between the GK domain and GPR domain through transferring unstable intermediate G5P (*Pérez-Arellano et al., 2010*; *Seddon et al., 1989*). Considering the distance between the GK and GPR domains is about 60 Å (*Figure 5—figure supplement 3*, *Figure 5—video 1*), we propose a model that P5CS filament may exhibit a scaffold architecture that stabilizes the relative position of the GK and GPR domains, the cooperation between which may produce electrostatic substrate channels that mediate the transfer of unstable intermediate G5P. In addition, P5CS filamentation may create a half-opened chamber with the active sites located at the inner part of the filament. Since the GK domain is catalytically faster than the GPR domain, the unstable intermediates G5P accumulate within the filament. This microenvironment may reduce the amount of G5P escaped into the solvent, thereby facilitating the rate-limiting reaction at the GPR domain.

## The working model

Together, we propose a coupling catalytic reaction mechanism of *Drosophila* dynamic P5CS filament. In this proposed model, spontaneous filamentation occurs at the APO state, and elongation of P5CS filament is associated with the binding of glutamate. Upon the binding of glutamate, the binding pocket at the GK domain is bound by ATP; subsequently, conformational changes facilitate the formation of a hook structure and phosphorylation of glutamate, which produces G5P. When products of the GK domain dissociate from the active site, G5P would be trapped by the channel and chamber within the filament and further captured by the GPR domain. Next, NADPH binds to the GPR domain, triggering the conformational change into closed conformation, which brings the NADPH towards the catalytic residue C598 and facilitates the reaction. After this reaction, NADP$^+$ and GSA will be released, and the GPR domain returns to its open conformation (*Figure 6*). This working model suggests that the GK and GPR domains undergo continuous conformational transition during catalysis, resulting in a dynamic filament.

In the P5CS$^{Glu}$ filament, the GK and GPR domains are likely in a stable conformational state, while vibration may occur in the GPR domain of P5CS$^{Glu/ATPγS}$ and P5CS$^{Mix}$ filaments due to the binding of ligands. This notion could be supported by the differences in their local resolution (*Figure 1B-D*). We speculate that the swing of GPR in the catalytic reaction could destabilize the interaction between adjacent GPR domain dimers in the filament. Therefore, the extra interaction at the hook structure of the GK domain may be required for the stabilization of the filament. This proposed stabilization is consistent with negative stain data showing that the P5CS$^{R124A}$ mutant cannot stabilize the filament structure in the catalytic process and lose the ability to form the long filaments.

## P5CS and human disease

Recently, accumulative evidences have shown that mutations on the human P5CS gene (*ALDH18A1*) is one of the causes of hyperammonemia, neurocutaneous syndromes, and motor neuron syndrome (*Baumgartner et al., 2005*; *Magini et al., 2019*; *Marco-Marín et al., 2020*; *Pérez-Arellano et al., 2010*). Such mutations may result in the loss of P5CS function in various degrees. *Drosophila* P5CS residue R124 in region I of the GK domain, which corresponds to R138 of human P5CS, is highly conserved among different eukaryotes (*Figure 5—figure supplement 4*). However, 17 residues in

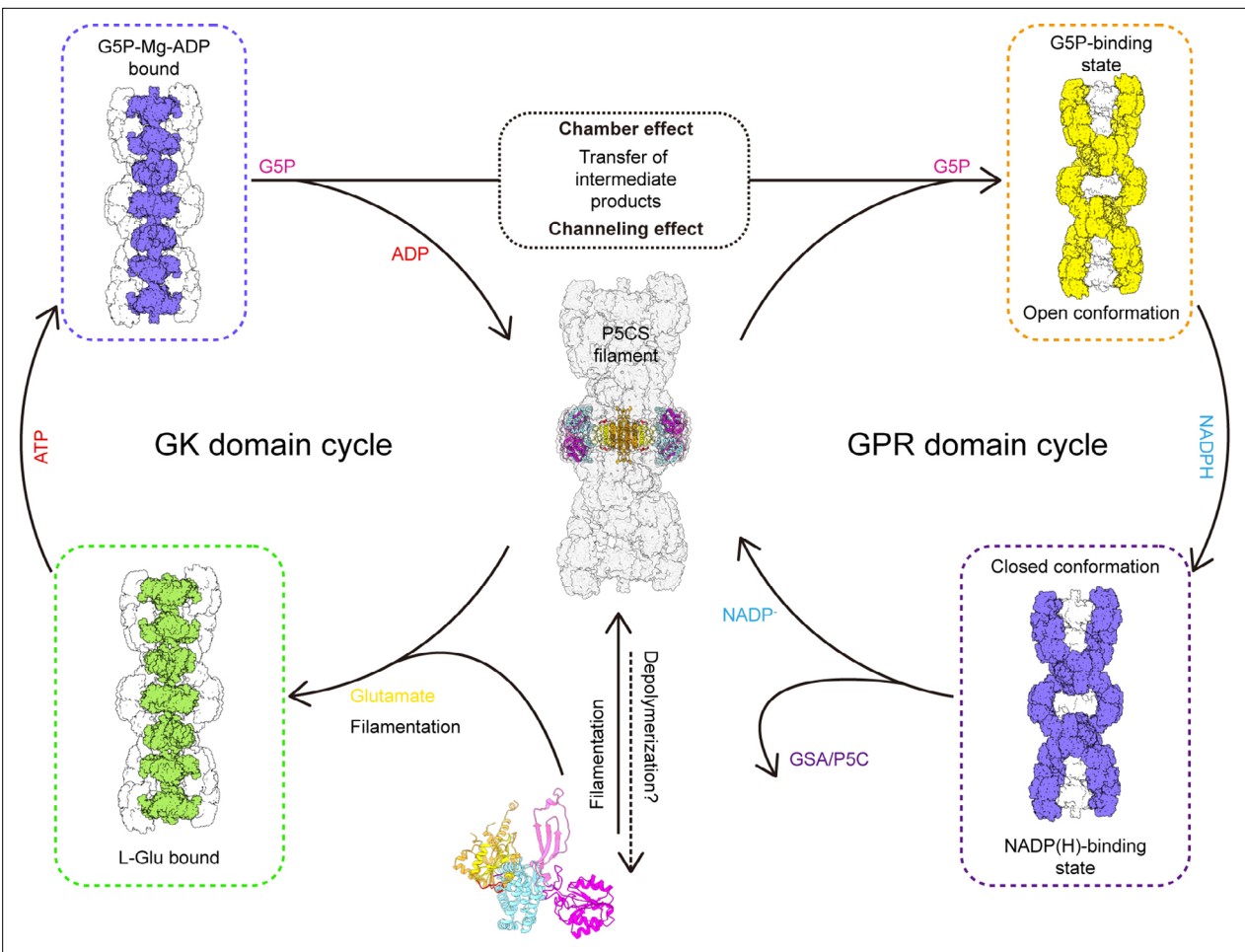

**Figure 6.** Model of P5CS filament structural transitions during GSA/P5C synthesis. The P5CS molecule polymerizes into filaments at the APO state or after binding with the glutamate. Upon ATP binding, the glutamate kinase (GK) domain initiates glutamate phosphorylation. The product leaves the pocket, and the GK domain subsequently repeats reaction cycle (left). Unstable G5P will be transported through channel and the half-open chamber inside the filament, and captured by the γ-glutamyl phosphate reductase (GPR) domain. NADPH binding to the GPR domain transforms the domain to closed conformation, which enables NADPH to approach the catalytic site and completes reductive dephosphorylation of G5P. The GSA/P5C will be released, and the GPR domain returns to the unliganded state with open conformation. The GPR domain then begins the next cycle (right).

region I, including the hook structure, are absent in *E. coli* GK. Pathogenic mutations of R138 in human P5CS, which are proven to be the cause of autosomal-dominant cutis laxa, have been demonstrated with a decreased activity and a dispersed distribution in mitochondria (*Fischer-Zirnsak et al., 2015*; *Yang et al., 2021*). In the protein structure database, there is only the GPR domain structure available for human P5CS (PDB: 2H5G). Its overall structure is similar to the GPR domain of *Drosophila* P5CS (*Figure 4—figure supplement 2B*). Although it is still unknown whether human P5CS can form filament structure in vitro, it is reasonable to suspect that the filament-forming property is conserved between human and *Drosophila* P5CS based on their structural similarity. Our structure reveals that the R138 mutation on human P5CS could abrogate the interaction between hook structures of GK domains, and thereby destabilize the filament and coupling of reactions at the two domains.

In summary, our cryo-EM structures of *Drosophila* P5CS filament present the assembly mode of P5CS protein and provide a molecular basis for a further understanding of the reaction mechanism of the GK and GPR domains. In our proposed model, the coupling of the GK and GPR domains in the filamentous structure facilitates the catalytic reaction of the bifunctional enzyme P5CS. Additional structural studies of P5CS filaments are required to determine whether there is an underlying regulatory mechanism that transmits information between the GK and GPR domains in the tetramer and along the filament.

# Materials and methods

## Key resources table

| Reagent type (species) or resource | Designation | Source or reference | Identifiers | Additional information |
|---|---|---|---|---|
| Gene (*Drosophila melanogaster*) | P5CS | GenBank | NM_001259948 | |
| Strain, strain background (*Escherichia coli*) | Transetta (DE3) | TransGen Biotech | | |
| Recombinant DNA reagent | pET28a-6His-SUMO | In house | | |
| Commercial assay or kit | BCA Protein Concentration Determination Kit (Enhanced) | Beyotime | P0010 | |
| Chemical compound, drug | Benzamidine hydrochloride | Sigma-Aldrich | 434760-5G | |
| Chemical compound, drug | Pepstatin A | Sigma-Aldrich | P5318-25MG | |
| Chemical compound, drug | Leupeptin hydrochloride microbial | Sigma/Aldrich | L9783-100MG | |
| Chemical compound, drug | PMSF | MDBio | P006-5g | |
| Chemical compound, drug | Ni-NTA Agarose | QIAGEN | 30250 | |
| Chemical compound, drug | L-Glutamic acid | Sigma-Aldrich | G1251-100G | |
| Chemical compound, drug | ATP | Takara | 4041 | |
| Chemical compound, drug | ATP-gamma-S | Abcam | ab138911 | |
| Chemical compound, drug | NADPH tetrasodium salt | Roche | 10107824001 | |
| Other | Nitinol mesh | Zhenjiang Lehua Electronic Technology | M024-Au300-R12/13 | Cryo-EM grid preparation |
| Other | Holey Carbon Film | Quantifoil | R1.2/1.3, 300 Mesh, Cu | Cryo-EM grid preparation |
| Other | 400 mesh reinforced carbon support film | EMCN | BZ31024a | Negative staining |
| Software, algorithm | UCSF Chimera | https://doi.org/10.1002/jcc.20084 | | https://www.cgl.ucsf.edu/chimera |
| Software, algorithm | UCSF ChimeraX | https://doi.org/10.1002/pro.3235 | | https://www.cgl.ucsf.edu/chimerax/ |
| Software, algorithm | RELION | https://doi.org/10.7554/eLife.42166 | | https://relion.readthedocs.io/en/latest/index.html# |
| Software, algorithm | Coot | https://doi.org/10.1107/S0907444910007493 | | https://www2.mrc-lmb.cam.ac.uk/personal/pemsley/coot/ |
| Software, algorithm | Phenix | https://doi.org/10.1107/S2059798318006551 | | https://phenix-online.org/ |

## P5CS protein purification

The full-length *D. melanogaster* P5CS gene was cloned into a modified pET28a vector with a 6 × His SUMO tag fused at the N terminus; the fusion proteins were expressed in *E. coli* Transetta (DE3) cells overnight at 16°C after induction with 0.1 mM IPTG at $OD_{600}$ range of 0.6–0.8. The remainder of purification was performed at 4°C. The harvested cells were sonicated under ice and purified by Ni-NTA agarose beads (QIAGEN) in lysis buffer (50 mM Tris-HCl pH 8.0, 500 mM NaCl, 10% glycerol, 20 mM imidazole, 1 mM PMSF, 5 mM β-mercaptoethanol, 5 mM benzamidine, 2 μg/ml leupeptin, and 2 μg/ml pepstatin). After in-column washing with lysis buffer, the proteins were eluted with elution buffer (50 mM Tris-HCl pH 8.0, 500 mM NaCl, 250 mM imidazole, 5 mM β-mercaptoethanol), peak fractions were treated with SUMO protease for 1 hr at 8°C. The P5CS proteins were further purified

through HiLoad 16/600 Superdex 200pg gel-filtration chromatography (GE Healthcare) in column buffer (25 mM HEPES pH 7.5 and 100 mM KCl), peak fractions were collected, concentrated, and stored at –80°C before use.

## Enzyme activity assays

The full-length wild-type or mutant P5CS (100 nM protein) activity was determined in the reaction buffer containing 25 mM HEPES pH 7.5, and 10 mM L-glutamate (Sigma), with added 20 mM $MgCl_2$, 10 mM ATP (Takara), and 0.5 mM NADPH (Roche) used to initiate the reaction (*Magini et al., 2019*; *Sabbioni et al., 2021*), then the reaction was monitored at 37°C in an MD-SpectraMax i3 plate reader and absorbance at 340 nm was measured every 20 s for 10 min (one experiment, n = 3). The NADPH concentration was converted from A340 with the standard curve determined at the same experiment.

## Negative staining

Wild-type or mutation P5CS proteins were mixed with different substrate conditions. In brief, the final concentration was as follows: 25 mM HEPES pH 7.5, 100 mM KCl, 10 mM $MgSO_4$, 100 mM L-glutamate, 10 mM ATP, and 0.5 mM NADPH. The prepared protein samples were applied to glow-discharged carbon-coated EM grids (400 mech, EMCN), and stained with 1% uranyl acetate. Negative-stain EM grids were photographed on a Tecnai Spirit G21 microscope (FEI).

## Cryo-EM grid preparation and data collection

For cryo-EM, purified full-length P5CS was diluted to approximately 2 µM and dissolved in buffer containing 25 mM HEPES pH 7.5, 100 mM KCl, 10 mM $MgSO_4$, and incubated with 20 mM L-glutamate for the P5CS$^{Glu}$ filament preparation. The P5CS$^{Glu/ATPγS}$ filament was added with an additional 0.5 mM ATPγS (Abcam) compared to the P5CS$^{Glu}$ filament. For the P5CS$^{Mix}$ filament, P5CS proteins (2 µM) were incubated with 100 mM KCl, 10 mM $MgSO_4$, 20 mM L-glutamate, 2 mM ATP, and 0.5 mM NADPH. All the samples were incubated for 1 hr on ice before vitrification. The P5CS filament samples were placed on $H_2/O_2$ glow-discharged holey carbon grids (Quantifoil Cu 300 mesh, R1.2/1.3) or amorphous alloy film (CryoMatrix M024-Au300-R12/13). Then, the grids were immediately blotted for 3.0 s and plunge-frozen in liquid ethane cooled by liquid nitrogen using Vitrobot (Thermo Fisher) at 4°C with 100% humidity. Images were collected on Titan Krios G3 (FEI) equipped with a K3 Summit direct electron detector (Gatan), operating in counting super-resolution mode at 300 kV with a total dose of 72 e⁻/Å², subdivided into 50 frames in 4 s exposure using SerialEM (*Mastronarde, 2005*). The images were recorded at a nominal magnification of 22,500 × and a calibrated pixel size of 1.06 Å, with defocus ranging from 0.8 to 2.5 µm.

## Image processing and 3D reconstruction

The whole-image analysis was performed with RELION3 (*Zivanov et al., 2018*). We used MotionCor2 (*Zheng et al., 2017*) and CTFFIND4 (*Rohou and Grigorieff, 2015*) via RELION GUI to pr-process the image, movie frames were aligned, and the contrast transfer function (CTF) parameters were estimated in this process. After manual selection, there are 4933 images for the P5CS$^{Glu}$ dataset, 6408 images for the P5CS$^{Glu/ATPγS}$ dataset, and 10,566 images for the P5CS$^{Mix}$ dataset left for further processing. For the flexibility of P5CS filaments, SPA was carried out in our reconstructions and no helical symmetry was implied in the whole process. Reference-free particle picking built in RELION3 was performed. This process provides 1,994,786 particles for P5CS$^{Glu}$, 2,024,372 particles for P5CS$^{Glu/ATPγS}$, and 8,027,582 particles for P5CS$^{Mix}$. At first, the particles were extracted binning two or three times for the fast 2D classifications. Datasets were cleaned with several rounds of 2D classification and the bin factors were gradually reduced to one at the same time. After 3D classifications with C1 symmetry were applied, several classes were selected to do finer 3D classifications with D2 symmetry. Classes with the intact structure were retained for 3D refinement with D2 symmetry. For the 3D refinement, 432,746, 327,841, and 1,412,498 particles were used for each dataset. The maps including three P5CS tetramer layers were obtained. The relative motion between GK and GPR limited the refinement at a high resolution, so we used the partition reconstruction strategy to improve the resolution for both the GK and GPR domains. For the GK domain, we used continued local refinement to improve the resolution with a mask focus on the middle layer GK. Then, the Ctf-refinement and Bayesian polishing were performed for the

remained particles and improved the resolution to 4. 1Å, 4.1 Å, and 3.6 Å for three-layer P5CS maps and 3.5 Å, 3.4 Å, and 3.1 Å for GK maps. For the GPR domain, particles were expended symmetry for the 3D classification without alignment. Several classes with the intact structure were selected and oriented; symmetry collapse was done at the same time. Then, 3D classifications and refinements with C2 symmetry were performed. For the P5CS$^{Mix}$, two different states of GPR were captured. Finally, we got 286,291, 348,804, 193,482, and 233,624 particles to construct maps for the GPR domain with 3.6 Å, 4.2 Å, 4.3 Å, and 4.2 Å resolutions. LocalRes was used to estimate the local resolution of our map.

### Model building refinement and validation

Based on our maps with the near-atomic resolution, the model of the GK and GPR domains was generated with focused refinement maps in different states. The initial model of the GK and GPR domains was generated via swiss model regarding 4Q1T (GK from *Burkholderia thailandensis*) and 2H5G (human GPR domain) as a reference, respectively. Manual adjustment and building the missing regions were done in Coot (*Emsley and Cowtan, 2004*). Real space refinements were performed with Phenix (*Adams et al., 2011*). The full-length P5CS models were linked using the corresponding GK and GPR structures; the linker was generated in the Coot and refined via Phenix. Figures and movies were generated with UCSF Chimera (*Pettersen et al., 2004*) and ChimeraX (*Goddard et al., 2018*).

## Acknowledgements

We thank Zhi-Jie Liu, Suwen Zhao, and Zherong Zhang for their helpful discussions. The EM data were collected at the ShanghaiTech Cryo-EM Imaging Facility. We also thank the Molecular and Cell Biology Core Facility (MCBCF) at the School of Life Science and Technology, ShanghaiTech University, for providing technical support. This work was supported by grants from the Ministry of Science and Technology of China (no. 2021YFA0804701-4), National Natural Science Foundation of China (no. 31771490), and Shanghai Science and Technology Commission (no. 20JC1410500).

## Additional information

### Funding

| Funder | Grant reference number | Author |
| --- | --- | --- |
| Ministry of Science and Technology of the People's Republic of China | 2021YFA0804701-4 | Ji-Long Liu |
| National Natural Science Foundation of China | 31771490 | Ji-Long Liu |
| Shanghai Science and Technology Commission | 20JC1410500 | Ji-Long Liu |

The funders had no role in study design, data collection and interpretation, or the decision to submit the work for publication.

### Author contributions

Jiale Zhong, Conceptualization, Data curation, Formal analysis, Investigation, Methodology, Software, Validation, Visualization, Writing – original draft, Writing – review and editing; Chen-Jun Guo, Conceptualization, Data curation, Formal analysis, Investigation, Methodology, Software, Validation, Visualization, Writing – review and editing; Xian Zhou, Conceptualization, Data curation, Formal analysis, Investigation, Methodology, Validation, Writing – review and editing; Chia-Chun Chang, Conceptualization, Formal analysis, Investigation, Validation, Writing – review and editing; Boqi Yin, Tianyi Zhang, Data curation, Formal analysis, Investigation, Methodology, Validation; Huan-Huan Hu, Guang-Ming Lu, Data curation, Formal analysis, Investigation, Validation; Ji-Long Liu, Conceptualization, Funding acquisition, Project administration, Resources, Supervision, Validation, Writing – original draft, Writing – review and editing

## Author ORCIDs

Jiale Zhong http://orcid.org/0000-0001-5873-0450
Chen-Jun Guo http://orcid.org/0000-0001-5342-4761
Xian Zhou http://orcid.org/0000-0002-0000-2415
Chia-Chun Chang http://orcid.org/0000-0001-7942-6300
Boqi Yin http://orcid.org/0000-0003-3974-9820
Tianyi Zhang http://orcid.org/0000-0002-4632-6298
Huan-Huan Hu http://orcid.org/0000-0002-5997-530X
Guang-Ming Lu http://orcid.org/0000-0002-6607-2264
Ji-Long Liu http://orcid.org/0000-0002-4834-8554

## Decision letter and Author response

Decision letter https://doi.org/10.7554/eLife.76107.sa1
Author response https://doi.org/10.7554/eLife.76107.sa2

---

# Additional files

## Supplementary files

• Transparent reporting form

## Data availability

Atomic models generated in this study have been deposited at the PDB under the accession codes 7F5T, 7F5U, 7F5V, 7F5X, 7WX3, 7WX4, 7WXF, 7WXG, 7WXH, 7WXI. Cryo-EM maps deposited to EMDB as: EMD-31466, EMD-31467, EMD-31468, EMD-31469, EMD-32875, EMD-32876, EMD-32877, EMD-32878, EMD-32879, EMD-32880. Source Data files have been provided for Figure 5D.

The following datasets were generated:

| Author(s) | Year | Dataset title | Dataset URL | Database and Identifier |
|---|---|---|---|---|
| Zhong J, Guo CJ, Zhou X, Liu JL | 2021 | *Drosophila* P5CS filament with glutamate. | https://www.rcsb.org/structure/7F5T | RCSB Protein Data Bank, 7F5T |
| Zhong J, Guo CJ, Zhou X, Liu JL | 2021 | GK domain of *Drosophila* P5CS filament with glutamate | https://www.rcsb.org/structure/7F5X | RCSB Protein Data Bank, 7F5X |
| Zhong J, Guo CJ, Zhou X, Liu JL | 2022 | GPR domain of *Drosophila* P5CS filament with glutamate | https://www.ebi.ac.uk/emdb/EMD-32877 | Electron Microscopy Data Bank, EMD-32877 |
| Zhong J, Guo CJ, Zhou X, Liu JL | 2022 | GPR domain of *Drosophila* P5CS filament with glutamate | https://www.rcsb.org/structure/7WXF | RCSB Protein Data Bank, 7WXF |
| Zhong J, Guo CJ, Zhou X, Liu JL | 2021 | *Drosophila* P5CS filament with glutamate and ATPγS | https://www.rcsb.org/structure/7F5U | RCSB Protein Data Bank, 7F5U |
| Zhong J, Guo CJ, Zhou X, Liu JL | 2022 | GK domain of *Drosophila* P5CS filament with glutamate and ATPγS | https://www.rcsb.org/structure/7WX4 | RCSB Protein Data Bank, 7WX4 |
| Zhong J, Guo CJ, Zhou X, Liu JL | 2022 | GPR domain of *Drosophila* P5CS filament with glutamate and ATPγS | https://www.rcsb.org/structure/7WXI | RCSB Protein Data Bank, 7WXI |
| Zhong J, Guo CJ, Zhou X, Liu JL | 2021 | *Drosophila* P5CS filament with glutamate, ATP, and NADPH | https://www.rcsb.org/structure/7F5V | RCSB Protein Data Bank, 7F5V |
| Zhong J, Guo CJ, Zhou X, Liu JL | 2022 | GK domain of *Drosophila* P5CS filament with glutamate, ATP, and NADPH | https://www.rcsb.org/structure/7WX3 | RCSB Protein Data Bank, 7WX3 |

*Continued on next page*

*Continued*

| Author(s) | Year | Dataset title | Dataset URL | Database and Identifier |
|---|---|---|---|---|
| Zhong J, Guo CJ, Zhou X, Liu JL | 2022 | GPR domain closed form of *Drosophila* P5CS filament with glutamate, ATP, and NADPH | https://www.rcsb.org/structure/7WXG | RCSB Protein Data Bank, 7WXG |
| Zhong J, Guo CJ, Zhou X, Liu JL | 2022 | GPR domain open form of *Drosophila* P5CS filament with glutamate, ATP, and NADPH | https://www.rcsb.org/structure/7WXH | RCSB Protein Data Bank, 7WXH |
| Zhong J, Guo CJ, Zhou X, Liu JL | 2021 | *Drosophila* P5CS filament with glutamate | https://www.ebi.ac.uk/emdb/EMD-31466 | Electron Microscopy Data Bank, EMD-31466 |
| Zhong J, Guo CJ, Zhou X, Liu JL | 2021 | GK domain of *Drosophila* P5CS filament with glutamate | https://www.ebi.ac.uk/emdb/EMD-31469 | Electron Microscopy Data Bank, EMD-31469 |
| Zhong J, Guo CJ, Zhou X, Liu JL | 2021 | *Drosophila* P5CS filament with glutamate and ATPγS | https://www.ebi.ac.uk/emdb/EMD-31467 | Electron Microscopy Data Bank, EMD-31467 |
| Zhong J, Guo CJ, Zhou X, Liu JL | 2022 | GK domain of *Drosophila* P5CS filament with glutamate and ATPγS | https://www.ebi.ac.uk/emdb/EMD-32876 | Electron Microscopy Data Bank, EMD-32876 |
| Zhong J, Guo CJ, Zhou X, Liu JL | 2022 | GPR domain of *Drosophila* P5CS filament with glutamate and ATPγS | https://www.ebi.ac.uk/emdb/EMD-32880 | Electron Microscopy Data Bank, EMD-32880 |
| Zhong J, Guo CJ, Zhou X, Liu JL | 2021 | *Drosophila* P5CS filament with glutamate, ATP, and NADPH | https://www.ebi.ac.uk/emdb/EMD-31468 | Electron Microscopy Data Bank, EMD-31468 |
| Zhong J, Guo CJ, Zhou X, Liu JL | 2022 | GK domain of *Drosophila* P5CS filament with glutamate, ATP, and NADPH | https://www.ebi.ac.uk/emdb/EMD-32875 | Electron Microscopy Data Bank, EMD-32875 |
| Zhong J, Guo CJ, Zhou X, Liu JL | 2022 | GPR domain closed form of *Drosophila* P5CS filament with glutamate, ATP, and NADPH | https://www.ebi.ac.uk/emdb/EMD-32878 | Electron Microscopy Data Bank, EMD-32878 |
| Zhong J, Guo CJ, Zhou X, Liu JL | 2022 | GPR domain open form of *Drosophila* P5CS filament with glutamate, ATP, and NADPH | https://www.ebi.ac.uk/emdb/EMD-32879 | Electron Microscopy Data Bank, EMD-32879 |

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
