## [Editor Report]

This paper reports the cryo-EM structures of *Drosophila* P5CS, an enzyme important in amino acid metabolism. This group had previously described P5CS filaments in *Drosophila*, and here show how the filaments are assembled. The paper describes structural changes that occur upon the binding of substrates and reaction intermediates, making a strong case for a conformational cycle that involves some loop movements. Importantly, the work shows that these movements occur in the context of the assembled filament. Point mutants that block filament assembly have reduced catalytic rates, suggesting that a role of the filament is to increase enzyme activity.

---

## [Decision Letter]

**Decision letter after peer review:**

Thank you for submitting your article "Structural basis of dynamic P5CS filaments" for consideration by *eLife*. Your article has been reviewed by 4 peer reviewers, one of wom is a member of our Board of Reviewing Editors, and the evaluation has been overseen by José Faraldo-Gómez as the Senior Editor. All reviewers have opted to remain anonymous.

1) The major criticism of all three reviewers was that the paper needs major rewriting both to improve the presentation and to improve its clarity.

2) Significant questions were raised about the need for more mechanistic insight. This may require some additional experiments such as those suggested, and would also require a better discussion in the paper about mechanisms. Further, some claims may need to be tempered in the absence of a structure with bound ATP.

*Reviewer #1 (Recommendations for the authors):*

The writing would greatly benefit from a professional editor.

The paper uses "spiral" in many places when they really mean "helical". A spiral is defined as: winding in a continuous and gradually widening (or tightening) curve, either around a central point on a flat plane or about an axis so as to form a cone. In contrast, a helix has a fixed radius.

*Reviewer #2 (Recommendations for the authors):*

1. Most importantly, how does filamentation facilitate the reaction? Is it merely by local concentration effects, a.k.a. "channeling" between active sites via the filament interior? Or is it via concerted or coupled conformational changes within the filament? Is it possible that particular interactions within the filament may be required to stabilize particular conformations of the enzyme required for its activity? If so, then the filament also performs this function in facilitating the enzyme reaction. One could address this by investigating whether or not each active site is fully functional in the absence of filamentation. This could be accomplished using the non-filament forming mutant enzymes, and testing for the presence of the product of the first step separately from that of the second, rather than for only the product of both steps, as done in Figure 5. If the reaction intermediate is too unstable, an active site mutation in the second step active site could be utilized to stall the second reaction step and enable G5P to be measured.

2. None of the structures contain ATP. One contains ADP. Speculation on what ATP does to the structure should be tempered by this fact. Structures of ATP dependent enzymes generally show distinct structures when bound to ATP vs. ADP, although this may or may not be relevant here. Still, the qualification should be made and the language more careful to reflect this fact.

3. Movement of NADPH towards a C598 is mentioned, and C598 is mentioned to be important for enzyme activity. What does C598 do? Where is it located? A figure showing the relationship between C598 and NADPH in the two conformations would be useful, as well as a description of why the close approach of these moieties is important. C598 is shown in Figure 4B, though not discussed in the text pertaining to that figure, and it is close to G5P. Could it merely be involved in G5P binding?

4. Figure 4B, inset – the close proximity of D715 to G5P would appear to not be a favorable interaction. Are these mediated by any cations, or is the pKa of D715 known to be raised perhaps to be involved in the reaction mechanism?

5. The document needs professional language editing throughout to make the text easier to read, although the issues did not impede this reviewer from understanding the main points of the manuscript.

6. Legends of Figures 2 and 3 should explain what the dashed lines represent. From the text, these appear to be disordered residues, but that is not clear in viewing the figure alone.

7. Figure 2: the structure with Glu/ATPgammaS is not shown, yet the text (page 9 line 156) implies it should be. Is it not shown because its conformation is the same as that with G5P-Mg-ADP? Also, the text refers to P5CS^Mix, but the figure shows "G5P-Mg-ADP bound". I gather that these are the same, but this should be made clear in one or both of the places they are discussed (i.e. the text and the figure). The figure legend mentions P5CS^Mix as having G5P-Mg-ADP bound, but the figure itself could also indicate which structure is which using the nomenclature of the text (i.e. P5CS^Glu/ATPgammaS and P5CS^Mix).

8. Page 9, line 165 "L-glutamate is bound in a vertical way". This is not obvious in the figure. Also, it is difficult to see the L-glutamate with the color scheme chosen (also true of the ADP).

9. Page 9, line 170, "the loop shifts away from the top of the binding pocket (Figure 3A)". Which loop? Can it be identified in the figure?

10. Page 9, line 171, M213 is mentioned, but not shown in the figure (sticks of the side chain can be seen if one looks very closely, but the residue should be labeled in the figure if it is mentioned in the text). What is the significance of M213? Why mention it?

11. Page 11, line 207, "G5P-binging", is this "G5P-binding"?

12. Page 12, line 221 and Figure 4F, why is it a "Rossmann-like fold" and not a "Rossman fold"?

13. Page 12, line 228, beta9-beta10 is mentioned in Figure 4F and Figure 4-Video 1, but not identified in the figure or video. Labeling the CD would also be helpful to connect to the text.

14. page 15, line 285. "First, the intermediate G5P is instability and the distance between the two reaction centers is about 60 Å". Does this mean G5P is unstable? How unstable? What is its lifetime under normal enzymatic conditions? A source for this information should also be cited.

15. Figure 5, S2 – it would be helpful to show roughly where the filament axis is in panel B – there is some indication in A, but the map is truncated making it not as obvious to the reader without additional guesswork. This would help to see how the filament may sequester to some degree G5P. Also, a view from the top may be helpful to identify spaces within the filament and where pores are located.

16. Page 16, line 307, "When the products of the GK domain dissociate from the pocket, the G5P is trapped within the filament and further captured by the GPR domain. Next, NADPH binds to the GPR domain, the conformational change brings the NADPH towards the catalytic residue C598, becoming the close conformation and facilitating the reaction". Is there evidence that NADPH does not bind until G5P is released? "close conformation" does this mean "closed conformation" or is it referring to the close proximity of C598? Again, the reader does not know the significance of C598 in how it participates in the reaction and this would be helpful to know here.

17. Page 16, line 319. Please add references to the statement in the first sentence of the paragraph regarding mutations and diseases.

18. Page 19, Enzyme assays. Please provide a reference for the method.

19. This reviewer is not a cryo-electron microscopist, so cannot comment on the quality of the data or its processing. However, is there a means to assess map/model agreement? For example, in crystallography one has the Rfree (in addition to quality indicators of the model geometry and x-ray data scaling, which has similar types of measures here) to determine the quality of the model. In cryoEM, perhaps a method such as cross-correlation and/or number of atoms within the envelope would be useful to assess how well the model and map agree?

20. Page 22, line 446, say what 4Q1T and 2h5g are (and why is one capitalized and one not?). It should be stated how these were used – was a homology model made with the P5CS sequence? How much of the model required changing?

21. Figure 3 and 4 – a color purple is indicated, but the structures in question appear blue.

22. Can the authors speculate on how ATP binding (really ADP) results in the formation of the hook? If not, can this be mentioned?

23. Page 12, Line 232, explain what ALDH family is and why this is relevant (presumably P5CS is a member, but this should be stated).

24. Why is are these new structures not analyzed and compared to the known crystal structure of human P5CS?

*Reviewer #3 (Recommendations for the authors):*

The weakness of the paper is the presentation in general. The authors should make an effort to improve the clarity of their descriptions and review the text carefully to correct grammatical issues.

This work reports high-quality cryo-EM reconstructions that reveal an impressive supramolecular filament of great beauty. My impression is that the description of the filament is correct, but that the structures offer a lot more of information that should allow to deepen in some crucial aspects such as the catalytic mechanisms, the communication of conformational changes or regulation between domains, which probably will be the subject of future publications.

My major criticism is that the manuscript requires a carefully re-writing to improve clarity and correct many language issues.

Some specific comments follow.

1. When describing the GK domain, the authors could reference the work describing the first G5K structure: Marco-Marín C, Gil-Ortiz F, Pérez-Arellano I, Cervera J, Fita I, Rubio V. A novel two-domain architecture within the amino acid kinase enzyme family revealed by the crystal structure of *Escherichia coli* glutamate 5-kinase. J Mol Biol. 2007 Apr 13;367(5):1431-46. doi: 10.1016/j.jmb.2007.01.073.

2. The G5K fold is described as "a sandwich-like α3β8α4 structure". Perhaps the authors could mention that this is fold is characteristic of the amino acid kinase family (http://pfam.xfam.org/family/PF00696), as predicted in: Ramón-Maiques S, Marina A, Gil-Ortiz F, Fita I, Rubio V. Structure of acetylglutamate kinase, a key enzyme for arginine biosynthesis and a prototype for the amino acid kinase enzyme family, during catalysis. Structure. 2002 Mar;10(3):329-42. doi: 10.1016/s0969-2126(02)00721-9.

3. When describing the structure of the GPR domain, the authors could mention that there is a crystal structure of the human P5CS GPR domain available in the PDB (entry 2H5G) without an accompanying publication.

4. It is unclear what is the ligand content for each of the reported cryo-EM reconstructions. The filaments grown with glutamate and the non-hydrolyzable ATP analog (ATPgS) showed the product glutamate 5-phosphate in the GPR active site. But what is the content of the GK domain in this structure? The authors state that (Page 9. Line 160) "Three conformations of the GK domain with the different ligands were revealed clearly in our models". However, only two structures (bound to glutamate or in complex with glutamate 5-phosphate, ADP and Mg) are described and represented in the figures.

The authors could consider adding images with the densities for the bound glutamate 5-phosphate, ADP and Mg (as they have already done for the glutamate 5-phosphate and for NADPH)

Also, table 1 uses acronyms (GGL RGP NAP) for the bound ligands, but their meaning is not explained. The table also lists no ions bound in any structure, although the text mentions the presence of Mg^2+^.

5. (Page 10. Line 191-196) The authors refer to "ATP binding" and this is misleading because none of the structures has ATP bound.

6. They refer to four different states for the GPR domain: (Page 11, lines 199) "… four different binding modes of GPR domain". But the differences between the apo state and the NADP(H)-released state are not clear since both show an empty active site.

7. The work reports the structures of the GPR domain bound to glutamate 5-phosphate or to NADP. Perhaps the authors could combine these structures by superimposing the active sites and provide some additional details about the reaction mechanism. Perhaps a composed figure that shows the active site with the bound NADP and G5P could be illustrative. The description of the proposed catalytic mechanism (page 12, lines 233-235) is not clear.

8. The authors could discuss why in presence of glutamate and ATPgS, the GPR active site has glutamate 5-phosphate bound, whereas in the filament grown with all substrates (glutamate, ATP and NADPH), the GPR active site only shows NADP bound.

9. There is certain inconsistency regarding the conditions for the formation of P5CS filament formation. It is not clear whether filament formation requires glutamate or not. In a previous paper (Zhang 2020), the authors stated that "Purified P5CS in apo state could hardly form filaments" and that "Removing glutamate from solution almost abolished P5CS filament formation". However, the current work describes that "P5CS proteins self-assemble into filaments without the requirement of ligands" but also that "addition of substrates could enhance the length of filaments" and that "In consistent (sic) with our previous study, L-glutamate (…) is critical in promoting the formation and stability of P5CS filament". Perhaps the authors should explain more clearly whether glutamate is needed or not, the required concentration, and provide any additional variables that influence the formation of the filaments in vitro.

10. (Page 15, lines 301-303) "In the proposed model, spontaneous filamentation and elongation of P5CS is importantly associated with the binding of glutamate". To me, it is not clearly described in the manuscript how glutamate favors the formation or enhances the elongation of the P5CS filament. Perhaps the authors should reformulate this sentence or add a more detailed explanation.

11. It is not explained why ATP triggers the depolymerization of the filaments formed by mutant R124A, since according to the results, ATP favors the formation of the hook structures that glue the protein tetramers along the filament. One would expect that the nucleotide enhances rather than destabilizes the mutated filaments.

12. One important conclusion in the manuscript is that "The disruption of P5CS filament may result in uncoupled catalytic reactions of bifunctional P5CS and a reduced activity". The authors measured the complete two-step reaction in the WT and mutated proteins. The decreased activity could mean that in absence of filaments, the product of the GK reaction, is not properly channeled to the GPR domain. However, the decreased overall reaction activity could also be caused by a reduced G5K efficiency. Perhaps the authors could compare the G5K activity of the WT and mutated proteins. If the efficiency of the partial reaction is similar, this would strongly support that defects in filament formation are causing a defect in the channeling of the intermediate metabolite to the GPR domain.

13. Some important references are missing. In addition to those indicated above (points 1 and 2), the authors should include references for RELION, COOT, Phenix, Chimera and for PDB entry 4Q1T cited in the methods.

*Reviewer #4 (Recommendations for the authors):*

A detailed list of issues and questions follows here.

Figure 1 suppl 1 – scalebars should be included on all micrographs; panel 1B should include all 3 ligand states evaluated in the paper.

Figure 1 suppl 2 – it is hard to know which regions of the filaments are covered by the masks. Please show the mask superimposed on the filament structure.

Page 7 – the discussion of focused classification is a bit confusing. Does "multiple different conformational states" mean multiple conformations within each liganded state, or that each liganded state had a unique conformation. I think it's the later, but if the former then this should be explained and a supplemental figure showing the refinement classification strategy should be presented.

Page 8 – in comparison to existing crystal structures, it would be helpful to know the % identity to the *E. coli* enzyme, and the RMSD when the two structures are aligned (and, ideally, a supplemental image showing superposition of the structures). Similarly, there appears to be a structure of part of the human enzyme available, pdb id 2h5g, and it would be good to know how similar this structure is to the reported *Drosophila* structure.

Page 9 – disordered loop in region II – what is meant by "opening" and "closure" in terms of the disordered region. It is difficult to see in Figure 2 A/B, but it looks like the ordered region common to both structures is pretty much the same? If there are differences, this could perhaps be displayed differently.

Page 9 – "with the binding of different ligands" – does this mean that the ligands induce different conformational states, or that the same three conformations were observed regardless of ligand state? – Also, what are the "three conformational states"? This paragraph describes two?

Figure 2 A/B – very hard to assess the similarities and differences between the structures here. At a minimum the same regions should be shown for both structures. But from this it looks like the ordered part of the Glu-bound structure is the same as the "mix" structure, and the disordered loops have just been drawn in different positions. Maybe a superposition of the models would help clarify the differences? – Ah, there is a superposition in Figure 3A. Here it is clear, with the exception of one (or two?) residues if the Glu structure that point up (Figure 3A, left-hand side of the region II loop) the structures are the same. Is there additional evidence (weak density, perhaps?) to support where the disordered loops have been drawn? Showing the quality of the em density in this region is important to judge the conclusions being drawn about the potential movement of this loop.

Page 9, Figure 2 C/D – it is unclear where the "closure loop" is here, can this be highlighted?

Page 9 – How was the conversion of substrate to phosphorylated intermediate (G5P) assessed? Is this based just on the fit to the density, or is there orthogonal evidence (mass spec or something)? It's hard to judge the quality of the fit into the partial map shown in Figure 2D – a supplemental figure with the region around the G5P/ADP showing the quality of the model in this region and demonstrating a better fit of G5P/ADP than Glu/ATP would be helpful. Or a comparison to the Glu/ATPgS structure might be convincing.

Figure 3C – A supplementary figure showing the fit of the atomic model for the "hook" into density in Region 1 would be helpful in assessing the conformational change modeled here.

Page 10, last paragraph – the residues contacting the ligands are all in nearly the same positions in the two structures. Conformational changes described above are distal to the binding site.

Figure 5 supplement 1A – the quality of the negative stain images of F642A is insufficient to assess the protein quality. One cannot discern whether the protein is in a monomeric or tetrameric state, and the apo state micrograph would suggest that aggregation may be a problem for this mutant. Either better stain images or orthogonal data (circular dichroism, melting curve, etc.) are required to be certain that the protein is folded and stable upon introduction of the Phe to Ala mutation.

Figure 5 supplement 1B – While it appears that ATP does limit polymerization of the R124A mutant, a more quantitative measurement would be helpful, especially in interpretation of the enzyme activity data in Figure 5D. I suggest light scattering or ultracentrifugation could be used to quantify the fraction of enzyme in polymers.

Page 14, first paragraph – It is unclear why the R124A mutation would destabilize polymers. The P5CS(Glu) structure (Figure 1B) shows that the polymers are stable in the absence of longitudinal "hook" interactions that are presumably disrupted by this mutation.

Figure 5D – enzyme assays. The methods should indicate what concentration enzyme was used in this assay, and what concentration was imaged in negative stain. A major question is whether filaments are observed for the wildtype protein at whatever concentration was used for the enzyme assay. Presumably the answer is yes, but if this is not shown it would bring into question what causes the effect of the mutations on activity.

These data should be quantified by calculating specific activity of the enzymes under these conditions, which would allow comparison of these data with published values for P5CS.

This assay reads out the second step in a two-step reaction mechanism. If the function of filaments is to couple the two activities as asserted in the text (see comment below), then one would also expect that the point mutants would affect the rate of the first reaction. This should be tested using an assay to monitor ATP hydrolysis in the first step. If the rates of both reactions are reduced by the mutants this would be consistent with a coupling mechanism of the filament, but if only the second step is affected this would be consistent with the authors' hypothesis that the filament increases local intermediate concentration.

Page 15 – The rationale behind "coupling" of catalytic reactions by filament assembly is not clear. As the enzyme appears to undergo a complete ligand binding and catalytic cycle in the context of the filament, it is not clear how the filament contacts are coupling activities. While I realize it is asking a lot to add another cryo-EM structure, it seems that the structure of free P5CS tetramers would be an important piece of data to have in interpreting how filaments might be increasing activity.

Is there evidence to support an internal channel for G5P, as suggested here (at least I assume that is what is meant by "electrostatic channeling")? If there were a relevant channel stabilized by the filament, it should be seen in the wildtype enzyme filament structure.

The proposal that filaments function to create a locally high concentration of intermediates is interesting, but should be tested. One way to do this would be to monitor NADPH production in the presence of G5P as a substrate – at high G5P concentrations one would expect the mutant protein to have the same enzymatic rate as the wildtype.

At the bottom of page 15, in proposing the model for "filament catalysis" the statement that upon dissociation from the active site G5P is "trapped within the filament" is not well supported. The architecture of the filament is such that G5 would appear likely to be able to freely diffuse away from the filament.

Figure 5, supplement 3 – It would be helpful to include subdomain delineation and the locations of regions I, II, and III with the sequence alignment.

Page 16/17 – The potential link between filament assembly and human disease is intriguing. Is there evidence that the human enzyme forms filaments? It would be good to indicate how well-conserved the filament assembly interfaces are between human and *Drosophila*.

In a similar vein, the introduction mentions the role of P5CS in plants being of potential significance in agriculture. It would be good to indicate how well conserved filament assembly interfaces are in the plant sequences, and whether based on that one would anticipate that they assemble filaments similar to the *Drosophila* structure reported here.

---

## [Author Response]

Reviewer #1 (Recommendations for the authors):The writing would greatly benefit from a professional editor.

We have rewritten the paper to improve the clarity and the presentation.

The paper uses "spiral" in many places when they really mean "helical". A spiral is defined as: winding in a continuous and gradually widening (or tightening) curve, either around a central point on a flat plane or about an axis so as to form a cone. In contrast, a helix has a fixed radius.

Thanks for the explanation. We replaced “spiral” with “helical”.

Reviewer #2 (Recommendations for the authors):1. Most importantly, how does filamentation facilitate the reaction? Is it merely by local concentration effects, a.k.a. "channeling" between active sites via the filament interior? Or is it via concerted or coupled conformational changes within the filament? Is it possible that particular interactions within the filament may be required to stabilize particular conformations of the enzyme required for its activity? If so, then the filament also performs this function in facilitating the enzyme reaction. One could address this by investigating whether or not each active site is fully functional in the absence of filamentation. This could be accomplished using the non-filament forming mutant enzymes, and testing for the presence of the product of the first step separately from that of the second, rather than for only the product of both steps, as done in Figure 5. If the reaction intermediate is too unstable, an active site mutation in the second step active site could be utilized to stall the second reaction step and enable G5P to be measured.

By analyzing the arrangement of P5CS tetramers inside the filament, we believe that one role of filamentation is to facilitate the enzyme reaction. Having screened a large number of mutations, we were unable to obtain stable non-filament forming mutants.

2. None of the structures contain ATP. One contains ADP. Speculation on what ATP does to the structure should be tempered by this fact. Structures of ATP dependent enzymes generally show distinct structures when bound to ATP vs. ADP, although this may or may not be relevant here. Still, the qualification should be made and the language more careful to reflect this fact.

We temper our claims in the absence of an ATP-bound structure.

3. Movement of NADPH towards a C598 is mentioned, and C598 is mentioned to be important for enzyme activity. What does C598 do? Where is it located? A figure showing the relationship between C598 and NADPH in the two conformations would be useful, as well as a description of why the close approach of these moieties is important. C598 is shown in Figure 4B, though not discussed in the text pertaining to that figure, and it is close to G5P. Could it merely be involved in G5P binding?

We add text at lines 228-237 and a supplementary figure (Figure 4—figure supplement 3), highlighting the role and location of C598:

“The GPR domain of P5CS belongs to aldehyde dehydrogenase (ALDH) superfamily. ALDHs family uses NAD(P)^+^ to catalyze the conversion of various aldehydes into their corresponding carboxylic acids. Many studies on ALDHs have shown that a conserved residue cysteine acts as the active site of nucleophile, forming thiohemiacetal intermediate with substrate (Koppaka et al., 2012; Liu et al., 1997; Perozich et al., 1999). Curiously, the NADPH-utilizing GPR domain of P5CS catalyzes the reverse reaction of ALDHs. ”

4. Figure 4B, inset – the close proximity of D715 to G5P would appear to not be a favorable interaction. Are these mediated by any cations, or is the pKa of D715 known to be raised perhaps to be involved in the reaction mechanism?

Residue D715 is highly conserved and close to the active site, and it is not clear what role D715 plays in GPR domain reaction. It might be involved in the reaction mechanism.

5. The document needs professional language editing throughout to make the text easier to read, although the issues did not impede this reviewer from understanding the main points of the manuscript.

We have rewritten the paper to improve the clarity and the presentation.

6. Legends of Figures 2 and 3 should explain what the dashed lines represent. From the text, these appear to be disordered residues, but that is not clear in viewing the figure alone.

We have revised Figure legends of Figures 2 and 3:

“The dashed lines represent disordered segments in this model.”

“On the left panel, the dashed lines in the model represent the open-loop (blue-violet) and closed-loop (green) in region II.”

7. Figure 2: the structure with Glu/ATPgammaS is not shown, yet the text (page 9 line 156) implies it should be. Is it not shown because its conformation is the same as that with G5P-Mg-ADP? Also, the text refers to P5CS^Mix, but the figure shows "G5P-Mg-ADP bound". I gather that these are the same, but this should be made clear in one or both of the places they are discussed (i.e. the text and the figure). The figure legend mentions P5CS^Mix as having G5P-Mg-ADP bound, but the figure itself could also indicate which structure is which using the nomenclature of the text (i.e. P5CS^Glu/ATPgammaS and P5CS^Mix).

We add text at lines 162-169 and a supplementary figure (Figure 2—figure supplement 1C, D) to depict the GK domain structure of the P5CS^Glu/ATPγS^ filament:

“We obtained a structure of the GK domain with the binding of glutamate in the P5CS^Glu^ filament (Figure 2A) and a second structure of the GK domain with G5P-Mg-ADP in the P5CS^Mix^ filament (Figure 2B and Figure 2—figure supplement 1B). In the P5CS^Glu/ATPγS^ filament, the ligands could not be determined due to incomplete densities (Figure 2—figure supplement 1C). The GK domain structure of the P5CS^Glu/ATPγS^ filament is virtually identical to that of the P5CS^Mix^ filament (Figure 2—figure supplement 1D).”

The ligand-bound states of the GK domain correspond to ligand-bound states of P5CS filament were indicated in Figure 2.

8. Page 9, line 165 "L-glutamate is bound in a vertical way". This is not obvious in the figure. Also, it is difficult to see the L-glutamate with the color scheme chosen (also true of the ADP).

We have added a supplementary figure (Figure 2—figure supplement 1D) with a different angle to highlight the position of glutamate. We have selected an outline font for a clear display.

9. Page 9, line 170, "the loop shifts away from the top of the binding pocket (Figure 3A)". Which loop? Can it be identified in the figure?

We have changed “the loop” into “a disordered segment” and highlighted it in the figure (Figure 3A, Figure 3—figure supplement 1A-D).

10. Page 9, line 171, M213 is mentioned, but not shown in the figure (sticks of the side chain can be seen if one looks very closely, but the residue should be labeled in the figure if it is mentioned in the text). What is the significance of M213? Why mention it?

We added text and a supplementary figure (Figure 3A, Figure 3—figure supplement 1E), highlighting the role and location of M213.

11. Page 11, line 207, "G5P-binging", is this "G5P-binding"?

Yes, it should be “G5P-binding”, we have revised it.

12. Page 12, line 221 and Figure 4F, why is it a "Rossmann-like fold" and not a "Rossman fold"?

The classical Rossmann fold contains six β-strands, whereas Rossmann-like folds contain only five β-strands, just like the GPR domain in figure 4F.

13. Page 12, line 228, beta9-beta10 is mentioned in Figure 4F and Figure 4-Video 1, but not identified in the figure or video. Labeling the CD would also be helpful to connect to the text.

We revised the description about NBD rotation at lines 272-275 and added a supplementary figure (Figure 4—figure supplement 3) to show the cylinder axis:

“Meanwhile, the entire NBD rotates approximately 15° along the cylinder axis (Figure 4—figure supplement 3A) and slides towards CD (Figure 4F; Figure 4-Video 1).”

14. page 15, line 285. "First, the intermediate G5P is instability and the distance between the two reaction centers is about 60 Å". Does this mean G5P is unstable? How unstable? What is its lifetime under normal enzymatic conditions? A source for this information should also be cited.

This means G5P is unstable; relevant references have been added.

15. Figure 5, S2 – it would be helpful to show roughly where the filament axis is in panel B – there is some indication in A, but the map is truncated making it not as obvious to the reader without additional guesswork. This would help to see how the filament may sequester to some degree G5P. Also, a view from the top may be helpful to identify spaces within the filament and where pores are located.

We have added a video (Figure 5-Video 1) and a view from the top (Figure 5—figure supplement 3) for a better representation.

16. Page 16, line 307, "When the products of the GK domain dissociate from the pocket, the G5P is trapped within the filament and further captured by the GPR domain. Next, NADPH binds to the GPR domain, the conformational change brings the NADPH towards the catalytic residue C598, becoming the close conformation and facilitating the reaction". Is there evidence that NADPH does not bind until G5P is released? "close conformation" does this mean "closed conformation" or is it referring to the close proximity of C598? Again, the reader does not know the significance of C598 in how it participates in the reaction and this would be helpful to know here.

This means “closed conformation”. We have added supplementary figures (Figure 4—figure supplement 1, Figure 4—figure supplement 2), highlighting the role and location of C598. We have expanded the discussion of a catalytic model of the GPR domain at lines 359-365:

“In the GPR domain of *Drosophila* P5CS, our data suggest that the catalytic residue C598 of CD attacks the G5P to form the first tetrahedral thioacetal intermediate in the reaction, and then expulsion of phosphate collapses to form a stable thioacyl enzyme intermediate. A hydride is then transferred to this intermediate from NADPH, with subsequent collapse to release the product GSA.”

17. Page 16, line 319. Please add references to the statement in the first sentence of the paragraph regarding mutations and diseases.

Relevant references have been added.

18. Page 19, Enzyme assays. Please provide a reference for the method.

Relevant references have been added

19. This reviewer is not a cryo-electron microscopist, so cannot comment on the quality of the data or its processing. However, is there a means to assess map/model agreement? For example, in crystallography one has the Rfree (in addition to quality indicators of the model geometry and x-ray data scaling, which has similar types of measures here) to determine the quality of the model. In cryoEM, perhaps a method such as cross-correlation and/or number of atoms within the envelope would be useful to assess how well the model and map agree?

To assess the agreement of map and model, the cross-correlation can be found in the PDB validation reports. As for the validation of the map, it is achieved by the gold standard FSC curve which is in Figure 1—figure supplement 2-4.

20. Page 22, line 446, say what 4Q1T and 2h5g are (and why is one capitalized and one not?). It should be stated how these were used – was a homology model made with the P5CS sequence? How much of the model required changing?

We use Swiss-model to build the initial model; 4Q1T (GK from *Burkholderia thailandensis*) and 2H5G (human GPR domain) were used as reference models to predict GK and GPR respectively. Then, we manually tuned each residue in Coot, using Phenix to refine and validate the final model.

21. Figure 3 and 4 – a color purple is indicated, but the structures in question appear blue.

Thank you for your suggestion. We have changed “purple” into “blue-violet”.

22. Can the authors speculate on how ATP binding (really ADP) results in the formation of the hook? If not, can this be mentioned?

We have expanded the discussion of nucleotides binding resulting in the formation of the hook structure at lines 335-346:

“We observed two ligand binding modes in the GK domain. Due to the lack of ATP-bound structure, it is difficult to determine whether ATP plays a decisive role in these conformational changes. According to a previous report on the N-Acetyl-L-glutamate kinase (NAGK), nucleoside is important for the conformational change of the AAK domain, and the structures are similar when bound by ADP or AMPPNP (Gil-Ortiz et al., 2011). Based on the similarity of sequences and structures between GK and NAGK (Marco-Marin et al., 2007), we propose that the conformation of the GK domain in the P5CS^Glu^ filament would transform upon the binding of ATP, thereby triggering the formation of hook structure and completing the catalytic reaction.”

23. Page 12, Line 232, explain what ALDH family is and why this is relevant (presumably P5CS is a member, but this should be stated).

Yes, P5CS is a member of the ALDH family. We have added text explaining the ALDH family.

24. Why is are these new structures not analyzed and compared to the known crystal structure of human P5CS?

We have added text and a supplementary figure (Figure 4—figure supplement 2B) to compare the structure of P5CS in *Drosophila* and that in human.

Reviewer #3 (Recommendations for the authors):The weakness of the paper is the presentation in general. The authors should make an effort to improve the clarity of their descriptions and review the text carefully to correct grammatical issues.This work reports high-quality cryo-EM reconstructions that reveal an impressive supramolecular filament of great beauty. My impression is that the description of the filament is correct, but that the structures offer a lot more of information that should allow to deepen in some crucial aspects such as the catalytic mechanisms, the communication of conformational changes or regulation between domains, which probably will be the subject of future publications.My major criticism is that the manuscript requires a carefully re-writing to improve clarity and correct many language issues.

We have rewritten the paper to improve the clarity and the presentation.

Some specific comments follow.1. When describing the GK domain, the authors could reference the work describing the first G5K structure: Marco-Marín C, Gil-Ortiz F, Pérez-Arellano I, Cervera J, Fita I, Rubio V. A novel two-domain architecture within the amino acid kinase enzyme family revealed by the crystal structure of *Escherichia coli* glutamate 5-kinase. J Mol Biol. 2007 Apr 13;367(5):1431-46. doi: 10.1016/j.jmb.2007.01.073.

Relevant references have been added.

2. The G5K fold is described as "a sandwich-like α3β8α4 structure". Perhaps the authors could mention that this is fold is characteristic of the amino acid kinase family (http://pfam.xfam.org/family/PF00696), as predicted in: Ramón-Maiques S, Marina A, Gil-Ortiz F, Fita I, Rubio V. Structure of acetylglutamate kinase, a key enzyme for arginine biosynthesis and a prototype for the amino acid kinase enzyme family, during catalysis. Structure. 2002 Mar;10(3):329-42. doi: 10.1016/s0969-2126(02)00721-9.

We have added text at lines 152-158:

“The GK domain of *Drosophila* P5CS is conserved with the GK protein in *E. coli*. Alignments of sequences and structures indicate that their secondary structures are similar as both exhibit a sandwich-like α3β8α4 topological folding (Figure 2—figure supplement 1A), which is a characteristic of the amino acid kinase (AAK) family (Marco-Marin et al., 2007; Perez-Arellano et al., 2010; Ramon-Maiques et al., 2002).”

3. When describing the structure of the GPR domain, the authors could mention that there is a crystal structure of the human P5CS GPR domain available in the PDB (entry 2H5G) without an accompanying publication.

We have added text and a supplementary figure (Figure 4—figure supplement 2B) to compare the structure of P5CS in *Drosophila* and that in human.

4. It is unclear what is the ligand content for each of the reported cryo-EM reconstructions. The filaments grown with glutamate and the non-hydrolyzable ATP analog (ATPgS) showed the product glutamate 5-phosphate in the GPR active site. But what is the content of the GK domain in this structure? The authors state that (Page 9. Line 160) "Three conformations of the GK domain with the different ligands were revealed clearly in our models". However, only two structures (bound to glutamate or in complex with glutamate 5-phosphate, ADP and Mg) are described and represented in the figures.The authors could consider adding images with the densities for the bound glutamate 5-phosphate, ADP and Mg (as they have already done for the glutamate 5-phosphate and for NADPH)Also, table 1 uses acronyms (GGL RGP NAP) for the bound ligands, but their meaning is not explained. The table also lists no ions bound in any structure, although the text mentions the presence of Mg^2+^.

We add text at “lines 162-169” and a supplementary figure (Figure 2—figure supplement 1C, D) to depict the GK domain structure of the P5CS^Glu/ATPγS^ filament:

“We obtained a structure of the GK domain with the binding of glutamate in the P5CS^Glu^ filament (Figure 2A) and a second structure of the GK domain with G5P-Mg-ADP in the P5CS^Mix^ filament (Figure 2B and Figure 2—figure supplement 1B). In the P5CS^Glu/ATPγS^ filament, the ligands could not be determined due to incomplete densities (Figure 2—figure supplement 1C). The GK domain structure of the P5CS^Glu/ATPγS^ filament is virtually identical to that of the P5CS^Mix^ filament (Figure 2—figure supplement 1D).”

We added a supplementary figure (Figure 2—figure supplement 1B) to show the cryo-EM densities for the bound G5P-Mg-ADP in the GK domain of P5CS^Glu^ filament.

GGL, RGP, and NAP are component identifiers (3-letter code) which are consist of RCSB PDB, representing the bound ligands in our models. We also updated table 1 and added the Mg^2+^ ion to the list.

5. (Page 10. Line 191-196) The authors refer to "ATP binding" and this is misleading because none of the structures has ATP bound.

We have removed this paragraph and expanded the discussion of ATP bound at lines 335-346:

“We observed two ligand binding modes in the GK domain. Due to the lack of ATP-bound structure, it is difficult to determine whether ATP plays a decisive role in these conformational changes. According to a previous report on the N-Acetyl-L-glutamate kinase (NAGK), nucleoside is important for the conformational change of the AAK domain, and the structures are similar when bound by ADP or AMPPNP (Gil-Ortiz et al., 2011). Based on the similarity of sequences and structures between GK and NAGK (Marco-Marin et al., 2007), we propose that the conformation of the GK domain in the P5CS^Glu^ filament would transform upon the binding of ATP, thereby triggering the formation of hook structure and completing the catalytic reaction.”

6. They refer to four different states for the GPR domain: (Page 11, lines 199) "… four different binding modes of GPR domain". But the differences between the apo state and the NADP(H)-released state are not clear since both show an empty active site.

We add text at lines 253-255 and a supplementary figure (Figure 4—figure supplement 2) to compare the NADP(H)-released state and the APO state.

“The structure of NADP(H)-release state, which has no bound ligand, is identical to the unliganded state.”

7. The work reports the structures of the GPR domain bound to glutamate 5-phosphate or to NADP. Perhaps the authors could combine these structures by superimposing the active sites and provide some additional details about the reaction mechanism. Perhaps a composed figure that shows the active site with the bound NADP and G5P could be illustrative. The description of the proposed catalytic mechanism (page 12, lines 233-235) is not clear.

We have added a figure (Figure 4—figure supplement 3B), showing the active site with the bound NADP and G5P. The description of the proposed catalytic mechanism has been rewritten in Discussion at lines 359-365:

“In GPR domain of *Drosophila* P5CS, our data suggest that the catalytic residue C598 of CD attacks the G5P to form the first tetrahedral thioacetal intermediate in the reaction, and then expulsion of phosphate collapses to form a stable thioacyl enzyme intermediate. A hydride is then transferred to this intermediate from NADPH, with subsequent collapse to release the product GSA.”

8. The authors could discuss why in presence of glutamate and ATPgS, the GPR active site has glutamate 5-phosphate bound, whereas in the filament grown with all substrates (glutamate, ATP and NADPH), the GPR active site only shows NADP bound.

We have added text at lines 243-244 and supplementary figures (Figure 2—figure supplement 1B and Figure 4—figure supplement 1) to discuss this result:

“It might be a contamination of ATP, leading to the production of the substrate G5P.”

“Figure 4—figure supplement 1. Representative cryo-electron microscopy (cryo-EM) densities for the active site of the γ-glutamyl phosphate reductase (GPR) domain.

(A) Cryo-EM map quality of G5P ligand in the active site of GPR domain in P5CS^Glu/ATPγS^ filament. (B) Unmodeled densities in the active site of GPR domain at NADP(H)-binding state, which may be the reaction product: π or GSA/P5C.”

9. There is certain inconsistency regarding the conditions for the formation of P5CS filament formation. It is not clear whether filament formation requires glutamate or not. In a previous paper (Zhang 2020), the authors stated that "Purified P5CS in apo state could hardly form filaments" and that "Removing glutamate from solution almost abolished P5CS filament formation". However, the current work describes that "P5CS proteins self-assemble into filaments without the requirement of ligands" but also that "addition of substrates could enhance the length of filaments" and that "In consistent (sic) with our previous study, L-glutamate (…) is critical in promoting the formation and stability of P5CS filament". Perhaps the authors should explain more clearly whether glutamate is needed or not, the required concentration, and provide any additional variables that influence the formation of the filaments in vitro.

We have revised the main text at lines 99-110 in response to the comment:

“In our previous study, we found that *Drosophila* P5CS in the APO state is hard to form filaments at low concentrations (<0.05 μM). The addition of glutamate to the P5CS samples induces micron-scale filaments (Zhang et al., 2020). Here, we observe that increasing P5CS concentration (>1 μM) also promotes the formation of filaments in the APO state. Our results show that the P5CS proteins can be self-assembled into filaments without ligands, and adding substrates increases the length of filaments at the same concentration of the P5CS proteins. Consistent with our previous study, glutamate (a substrate of P5CS) promotes the formation and maintenance of *Drosophila* P5CS filaments (Zhang et al., 2020).”

10. (Page 15, lines 301-303) "In the proposed model, spontaneous filamentation and elongation of P5CS is importantly associated with the binding of glutamate". To me, it is not clearly described in the manuscript how glutamate favors the formation or enhances the elongation of the P5CS filament. Perhaps the authors should reformulate this sentence or add a more detailed explanation.

Revised as the following:

“In this proposed model, spontaneous filamentation occurs at APO state, and elongation of P5CS filament is associated with the binding of glutamate.”

We have expanded the discussion at lines 346-349 in response to the comment:

“Although we solved the clear structure of the P5CS^Glu^ filament, further research is needed to understand how the conformation of glutamate binding contributes to the extension of P5CS filaments.”

11. It is not explained why ATP triggers the depolymerization of the filaments formed by mutant R124A, since according to the results, ATP favors the formation of the hook structures that glue the protein tetramers along the filament. One would expect that the nucleotide enhances rather than destabilizes the mutated filaments.

We have added text and expanded the discussion of short *P5CS^R124A^* filaments at lines 423-430:

“We speculate that the swing of GPR in the catalytic reaction could destabilize the interaction between adjacent GPR domain dimers in the filament. Therefore, the extra interaction at the hook structure of the GK domain may be required for the stabilization of the filament. This proposed stabilization is consistent with negative stain data showing that the P5CS^R124A^ mutant cannot stabilize the filament structure in the catalytic process and lose the ability to form the long filaments.”

12. One important conclusion in the manuscript is that "The disruption of P5CS filament may result in uncoupled catalytic reactions of bifunctional P5CS and a reduced activity". The authors measured the complete two-step reaction in the WT and mutated proteins. The decreased activity could mean that in absence of filaments, the product of the GK reaction, is not properly channeled to the GPR domain. However, the decreased overall reaction activity could also be caused by a reduced G5K efficiency. Perhaps the authors could compare the G5K activity of the WT and mutated proteins. If the efficiency of the partial reaction is similar, this would strongly support that defects in filament formation are causing a defect in the channeling of the intermediate metabolite to the GPR domain.

We have tempered our claims and added a Discussion section for the potential working model.

13. Some important references are missing. In addition to those indicated above (points 1 and 2), the authors should include references for RELION, COOT, Phenix, Chimera and for PDB entry 4Q1T cited in the methods.

Relevant references have been added.

Reviewer #4 (Recommendations for the authors):In support of, and addition to, the broad comments included in the public review, a detailed list of issues and questions follows here.Figure 1 suppl 1 – scalebars should be included on all micrographs; panel 1B should include all 3 ligand states evaluated in the paper.

We have added scale bars to all micrographs, and cryo-EM micrographs of three ligand states are provided (Figure 2—figure supplement 1E-G).

Figure 1 suppl 2 – it is hard to know which regions of the filaments are covered by the masks. Please show the mask superimposed on the filament structure.

We have superimposed masks on corresponding filament structures in our flowchart

Page 7 – the discussion of focused classification is a bit confusing. Does "multiple different conformational states" mean multiple conformations within each liganded state, or that each liganded state had a unique conformation. I think it's the later, but if the former then this should be explained and a supplemental figure showing the refinement classification strategy should be presented.

It is the later. The word “different” has been removed at lines 121-124:

“Using separate focused refinement strategy, we obtained multiple conformational states of the GK domain tetramer (3.1 to 3.5 Å) and the GPR domain dimer (3.6 to 4.3 Å).”

Page 8 – in comparison to existing crystal structures, it would be helpful to know the % identity to the *E. coli* enzyme, and the RMSD when the two structures are aligned (and, ideally, a supplemental image showing superposition of the structures). Similarly, there appears to be a structure of part of the human enzyme available, pdb id 2h5g, and it would be good to know how similar this structure is to the reported *Drosophila* structure.

We have added text at lines 825-827 and a supplementary figure (Figure 2—figure supplement 1A) to show the comparison of *E. coli* GK and the GK domain of *Drosophila* P5CS:

“(A) The comparison of *E. coli* GK structure without PUA domain (tan; PDB: 2J5V) and *Drosophila* GK domain structure of P5CS (blue-violet; this study), with 30.56% sequence identity and RMSD value of 1.363 Å (198 atom pairs).”

We also add text at lines 464-449, 892-894, and a supplementary figure (Figure 4—figure supplement 2B) to compare the structure of P5CS in *Drosophila* and that in human:

“In the protein structure database, there is only the GPR domain structure available for human P5CS (PDB: 2H5G). Its overall structure is similar to the GPR domain of *Drosophila* P5CS (Figure 4—figure supplement 2B).”

“Compared with the GPR domain of *Drosophila* P5CS at NADP(H)-binding state (blue-violet), the GPR domain of human P5CS (cyan, PDB:2H5G) has 56.74% sequence identity and RMSD value of 1.531 Å (398 atom pairs).”

Page 9 – disordered loop in region II – what is meant by "opening" and "closure" in terms of the disordered region. It is difficult to see in Figure 2 A/B, but it looks like the ordered region common to both structures is pretty much the same? If there are differences, this could perhaps be displayed differently.

We have rewritten the description of the disordered region and suggested that they referred to open-loop and closed-loop at lines 190-202, which were shown in Figure 3A and Figure 3—figure supplement 1:

“Meanwhile, based on the disorder densities in region II (Figure 3—figure supplement 1A-D), we modeled the possible trend of the missing segment with a dashed line (Figure 3A). In the P5CS^Glu^ filament, we speculate that the disordered segment in region II acts as a closed-loop, which traps glutamate in GBD (Figure 2C, Figure 3A). In the P5CS^Mix^ filament, the same segment shifts away from the top of the binding pocket and forms an open-loop, in which residue M213 interacts with G5P (Figure 3A, Figure 3—figure supplement 1E). We notice that closed-loop has a steric clash with G5P, preventing the binding of G5P under such a conformation (Figure 3—figure supplement 1D). Our findings support the idea that region II at the GK domain engages in regulating the catalytic reaction.”

Page 9 – "with the binding of different ligands" – does this mean that the ligands induce different conformational states, or that the same three conformations were observed regardless of ligand state? – Also, what are the "three conformational states"? This paragraph describes two?

We have rewritten the main text at lines161-168, and added supplementary figures ( figure 2—figure supplement 1C, D) to show three conformational states of the GK domain:

“We obtained a structure of the GK domain with the binding of glutamate in the P5CS^Glu^ filament (Figure 2A) and a second structure of the GK domain with G5P-Mg-ADP in the P5CS^Mix^ filament (Figure 2B and Figure 2—figure supplement 1B). In the P5CS^Glu/ATPγS^ filament, the ligands could not be determined due to incomplete densities (Figure 2—figure supplement 1C). The GK domain structure of the P5CS^Glu/ATPγS^ filament is virtually identical to that of the P5CS^Mix^ filament (Figure 2—figure supplement 1D).”

Figure 2 A/B – very hard to assess the similarities and differences between the structures here. At a minimum the same regions should be shown for both structures. But from this it looks like the ordered part of the Glu-bound structure is the same as the "mix" structure, and the disordered loops have just been drawn in different positions. Maybe a superposition of the models would help clarify the differences? – Ah, there is a superposition in Figure 3A. Here it is clear, with the exception of one (or two?) residues if the Glu structure that point up (Figure 3A, left-hand side of the region II loop) the structures are the same. Is there additional evidence (weak density, perhaps?) to support where the disordered loops have been drawn? Showing the quality of the em density in this region is important to judge the conclusions being drawn about the potential movement of this loop.

We add text and supplementary figures (Figure 3A, Figure 3—figure supplement 1A-C) to show the quality of the EM density in region II.

Page 9, Figure 2 C/D – it is unclear where the "closure loop" is here, can this be highlighted?

We highlighted the “open-loop” and “closed-loop” in the Figure 2C and D.

Page 9 – How was the conversion of substrate to phosphorylated intermediate (G5P) assessed? Is this based just on the fit to the density, or is there orthogonal evidence (mass spec or something)? It's hard to judge the quality of the fit into the partial map shown in Figure 2D – a supplemental figure with the region around the G5P/ADP showing the quality of the model in this region and demonstrating a better fit of G5P/ADP than Glu/ATP would be helpful. Or a comparison to the Glu/ATPgS structure might be convincing.

We have added a supplementary figure (Figure 2—figure supplement 1B) to show that the binding modes of Glu/ATP and G5P/ADP ligands, indicating that the conformation of bound G5P/ADP is better than Glu/ATP.

Figure 3C – A supplementary figure showing the fit of the atomic model for the "hook" into density in Region 1 would be helpful in assessing the conformational change modeled here.

We have added a supplementary figure (Figure 3—figure supplement 1F) to show the conformational change of hook structure in region I, and cartoon models have overlaid the cryo-EM density of P5CS^Glu^ filament shown as mesh.

And hook structure fitting into density was shown in Figure 1—figure supplement 4C.

Page 10, last paragraph – the residues contacting the ligands are all in nearly the same positions in the two structures. Conformational changes described above are distal to the binding site.

We have rewritten the description of GK domain structural changes at lines 221-224:

“By comparing the structures of GK domain with various ligands, we demonstrate the conformational changes, which may be associated with phosphorylation of the substrate glutamate.”

Figure 5 supplement 1A – the quality of the negative stain images of F642A is insufficient to assess the protein quality. One cannot discern whether the protein is in a monomeric or tetrameric state, and the apo state micrograph would suggest that aggregation may be a problem for this mutant. Either better stain images or orthogonal data (circular dichroism, melting curve, etc.) are required to be certain that the protein is folded and stable upon introduction of the Phe to Ala mutation.

According to a large number of mutant screening and structural analysis, we speculate that *Drosophila* P5CS cannot form a stable free tetramer, or the tetramer form only exists in the filament. We selected the high concentration protein, which will indeed affect the quality of staining. We found that aggregation is indeed a problem of F642A mutant. In our purification process, this mutant showed very low activity and was easy to inactivate, and the irregular spherical protein and no regular monomer or tetramer were observed by TEM.

Figure 5 supplement 1B – While it appears that ATP does limit polymerization of the R124A mutant, a more quantitative measurement would be helpful, especially in interpretation of the enzyme activity data in Figure 5D. I suggest light scattering or ultracentrifugation could be used to quantify the fraction of enzyme in polymers.

We have rewritten the description of P5CS^R124A^ filaments at lines 316-321:

“In contrast, the P5CS^R124A^ mutant proteins formed long filaments in the APO state as well as in the presence glutamate (Figure 5—figure supplement 2B). We observed that glutamate-bound P5CS^R124A^ filaments disassembled at the initial phase of adding ATP. Being incubated with all substrates, P5CS^R124A^ formed shorter filaments than P5CS^WT^ (Figure 5—figure supplement 2B).”

For negative stain sample preparation, all substrates are excessive. The effect of ATP on the R124A mutant can only be captured in the initial phase of the GK domain reaction. With the extension of incubation time, a short filament will form eventually. Our data show that ATP can affect the R124A mutant, highlighting the importance of the hook structure.

Page 14, first paragraph – It is unclear why the R124A mutation would destabilize polymers. The P5CS(Glu) structure (Figure 1B) shows that the polymers are stable in the absence of longitudinal "hook" interactions that are presumably disrupted by this mutation.

We have added text and expanded the discussion of short *P5CS^R124A^* filaments at lines 423-430:

“We speculate that the swing of GPR in the catalytic reaction could destabilize the interaction between adjacent GPR domain dimers in the filament. Therefore, the extra interaction at the hook structure of the GK domain may be required for the stabilization of the filament. This proposed stabilization is consistent with negative stain data showing that the P5CS^R124A^ mutant cannot stabilize the filament structure in the catalytic process and lose the ability to form the long filaments.”

Figure 5D – enzyme assays. The methods should indicate what concentration enzyme was used in this assay, and what concentration was imaged in negative stain. A major question is whether filaments are observed for the wildtype protein at whatever concentration was used for the enzyme assay. Presumably the answer is yes, but if this is not shown it would bring into question what causes the effect of the mutations on activity.These data should be quantified by calculating specific activity of the enzymes under these conditions, which would allow comparison of these data with published values for P5CS.This assay reads out the second step in a two-step reaction mechanism. If the function of filaments is to couple the two activities as asserted in the text (see comment below), then one would also expect that the point mutants would affect the rate of the first reaction. This should be tested using an assay to monitor ATP hydrolysis in the first step

We used 100 nM protein in enzyme assays and revised in our Methods, this protein concentration actually can induce the formation of filaments. And all the protein concentrations are marked on negative stain micrographs.

Our aim for enzyme activity assays is to test whether the filament structure is necessary for its reaction. As for the mechanism, more studies are needed. From our results, we observed that the activity of the R124A mutant was greatly reduced. We speculate that it will not have much effect on the rate of the first reaction, because the residue is not close to the active site. If the rate of the first reaction decreases, the destruction of the hook structure may lead to the conformational change of the GK domain tetramer. We have expanded this discussion at lines 423-430:

“We speculate that the swing of GPR in the catalytic reaction could destabilize the interaction between adjacent GPR domain dimers in the filament. Therefore, the extra interaction at the hook structure of the GK domain may be required for the stabilization of the filament. This proposed stabilization is consistent with negative stain data showing that the P5CS^R124A^ mutant cannot stabilize the filament structure in the catalytic process and lose the ability to form the long filaments.”

If the rates of both reactions are reduced by the mutants this would be consistent with a coupling mechanism of the filament, but if only the second step is affected this would be consistent with the authors' hypothesis that the filament increases local intermediate concentration.Page 15 – The rationale behind "coupling" of catalytic reactions by filament assembly is not clear. As the enzyme appears to undergo a complete ligand binding and catalytic cycle in the context of the filament, it is not clear how the filament contacts are coupling activities. While I realize it is asking a lot to add another cryo-EM structure, it seems that the structure of free P5CS tetramers would be an important piece of data to have in interpreting how filaments might be increasing activity.Is there evidence to support an internal channel for G5P, as suggested here (at least I assume that is what is meant by "electrostatic channeling"? If there were a relevant channel stabilized by the filament, it should be seen in the wildtype enzyme filament structure.The proposal that filaments function to create a locally high concentration of intermediates is interesting, but should be tested. One way to do this would be to monitor NADPH production in the presence of G5P as a substrate – at high G5P concentrations one would expect the mutant protein to have the same enzymatic rate as the wildtype.At the bottom of page 15, in proposing the model for "filament catalysis" the statement that upon dissociation from the active site G5P is "trapped within the filament" is not well supported. The architecture of the filament is such that G5 would appear likely to be able to freely diffuse away from the filament.Figure 5, supplement 3 – It would be helpful to include subdomain delineation and the locations of regions I, II, and III with the sequence alignment.

We thank the reviewer for this suggestion and agree that is also a problem we want to solve in the future. We have tempered our claims and proposed a model for the discussion of the possible channel or chamber or communication between GK and GPR domains at lines 380-389:

“As mentioned in the ‘Results’ section, we observed that mutated residues R124A and F642A do not directly participate in the active sites, while they are crucial for filamentation. This suggests that the P5CS filamentation couples the reaction catalyzed between the GK domain and GPR domain through transferring unstable intermediate G5P(Pérez-Arellano et al., 2010; Seddon et al., 1989). Considering the distance between the GK and GPR domains is about 60 Å (Figure 5—figure supplement 3, Figure 5-Video 1Figure 5—video 1), we propose a model that P5CS filament may exhibit a scaffold architecture that stabilizes the relative position of the GK and GPR domains, the cooperation between which may produce electrostatic substrate channels that mediate the transfer of unstable intermediate G5P. In addition, P5CS filamentation may create a half-opened chamber with the active sites located at the inner part of the filament. Since the GK domain is catalytically faster than the GPR domain, the unstable intermediates G5P accumulate within the filament. This microenvironment may reduce the amount of G5P escaped into the solvent, thereby facilitating the rate-limiting reaction at the GPR domain.”

In our results, P5CS protein showed high filament-forming ability. The filamentation of P5CS is like a highly ordered factory to facilitate the two-step reaction, which is also an important feature of P5CS filament. Whether dynamic P5CS filaments produce transient conformations to promote the transfer of intermediates, resulting in the formation of the channel or chamber to couple filament reactions, more structural studies of P5CS filaments are required to determine an underlying regulatory mechanism that transmits information between the GK and GPR domain in the tetramer and along the filament.

We have updated this supplementary figure. Subdomain and these regions are labeled now.

Page 16/17 – The potential link between filament assembly and human disease is intriguing. Is there evidence that the human enzyme forms filaments? It would be good to indicate how well-conserved the filament assembly interfaces are between human and *Drosophila*.In a similar vein, the introduction mentions the role of P5CS in plants being of potential significance in agriculture. It would be good to indicate how well conserved filament assembly interfaces are in the plant sequences, and whether based on that one would anticipate that they assemble filaments similar to the *Drosophila* structure reported here.

We have added text at lines 445-452:

“In the protein structure database, there is only the GPR domain structure available for human P5CS (PDB: 2H5G). Its overall structure is similar to the GPR domain of *Drosophila* P5CS (Figure 4—figure supplement 2B).Although it is still unknown whether human P5CS can form filament structure in vitro, it is reasonable to suspect that the filament-forming property is conserved between human and *Drosophila* P5CS based on their structural similarity.”

In the species we compared, the sequence of the interfaces is relatively conservative in animals, and less conservative in *Arabidopsis*. According to the P5CS structure of *Arabidopsis* predicted by AlphaFold2, its secondary structure on the interfaces are similar to that of *Drosophila*.